# Sodium channels implement a molecular leaky integrator that detects action potentials and regulates neuronal firing

Marco A Navarro[1], Autoosa Salari[2], Jenna L Lin[1†], Luke M Cowan[1‡], Nicholas J Penington[3], Mirela Milescu[1], Lorin S Milescu[1,4*]

[1]Division of Biological Sciences, University of Missouri, Columbia, United States; [2]Department of Molecular and Cell Biology, University of California, Berkeley, Berkeley, United States; [3]Department of Physiology and Pharmacology, SUNY Downstate Health Sciences University, Brooklyn, United States; [4]Department of Biology, University of Maryland, College Park, United States

**Abstract** Voltage-gated sodium channels play a critical role in cellular excitability, amplifying small membrane depolarizations into action potentials. Interactions with auxiliary subunits and other factors modify the intrinsic kinetic mechanism to result in new molecular and cellular functionality. We show here that sodium channels can implement a molecular leaky integrator, where the input signal is the membrane potential and the output is the occupancy of a long-term inactivated state. Through this mechanism, sodium channels effectively measure the frequency of action potentials and convert it into $Na^+$ current availability. In turn, the $Na^+$ current can control neuronal firing frequency in a negative feedback loop. Consequently, neurons become less sensitive to changes in excitatory input and maintain a lower firing rate. We present these ideas in the context of rat serotonergic raphe neurons, which fire spontaneously at low frequency and provide critical neuromodulation to many autonomous and cognitive brain functions.

**\*For correspondence:**
LorinSMilescu@gmail.com

**Present address:** [†]Department of Neuroscience, University of Wisconsin, Madison, United States; [‡]Division of Gastroenterology and Hepatology, Mayo Clinic, Rochester, United States

**Competing interests:** The authors declare that no competing interests exist.

## Introduction

Computation in the brain begins at the molecular level, with proteins such as ion channels and receptors that can change their structural and functional state in response to changes in the environment. These molecular building blocks capable of processing information have been adapted by nature into progressively more complex computational structures: ion channels and receptors were incorporated into synapses and neurons, neurons were interconnected into networks and circuits, and circuits were assembled into a brain capable of abstract thinking. Not surprisingly, computation in the engineering world followed the same trend, from transistors and integrated circuits to microprocessors and computers.

At the molecular level, voltage-gated sodium (Nav) channels have long been credited with a critical role in cellular excitability: amplifying a small membrane depolarization, such as created by a tiny postsynaptic excitatory current, into a full blown action potential (*Hodgkin and Huxley, 1952*). Computationally, Nav channels can be regarded as the equivalent of a transistor (*Sigworth, 2003*), a nonlinear electric circuit element. To generate action potentials of specific shape and firing patterns, spontaneously or in response to synaptic input, a neuron expresses a complement of Nav and other types of ion channels (*Bean, 2007*) and positions them strategically at subcellular locations (*Kole and Stuart, 2012*). Although we generally understand how ion channels contribute to the mechanics of action potential generation and propagation, the molecular and cellular landscapes are complex and remain incompletely charted. At the most basic level, we do not fully understand how

ion channels function as molecular computational machines and how they interact with each other and with other factors to regulate cellular activity.

We examine here a new computational function of the Nav channel that emerges from a process of long-term inactivation (LTI), which can be caused by interaction with fibroblast growth factor-homologous factors (FHFs), a relatively recently discovered group of auxiliary factors (*Goldfarb, 2012*). We are particularly interested in this functionality in the context of pacemaker serotonergic raphe neurons (*Jacobs and Azmitia, 1992*), which provide critical neuromodulation to many brain areas involved in autonomous and cognitive functions. In a previous study (*Milescu et al., 2010b*), we examined the Nav channels in *raphe obscurus* (RO) neurons and proposed a kinetic model that explains not only their intrinsic kinetic properties but also their characteristic process of long-term inactivation. Here, we investigate the computational aspects of this mechanism, using a combination of electrophysiology experiments and mathematical analysis.

## Results

### Action potentials in serotonergic raphe neurons and the contribution of Nav channels

Action potentials vary in their properties in different neuronal types but they generally last from hundreds of microseconds to several milliseconds, rapidly swinging the membrane between hyperpolarized and depolarized states. Serotonergic neurons have a particular electrophysiological profile, characterized by regular and spontaneous spiking at low frequency (3 – 5 Hz), a steady depolarization in the interspike interval, and broad action potentials (3 – 6 ms), as illustrated in *Figure 1A and B*. These characteristics are partially shared with other monoaminergic neurons (*Grace and Bunney, 1983*; *Vandermaelen and Aghajanian, 1983*; *Li and Bayliss, 1998*; *de Oliveira et al., 2010*; *Tuckwell and Penington, 2014*).

As in most excitable cells, Nav channels play a central role in serotonergic neurons, releasing the large depolarizing $Na^+$ current ($I_{Na}$) that generates the action potential (*Milescu et al., 2010b*). To perform their duty, Nav channels must cycle through a sequence of functional states, as summarized in *Figure 1C*: they are (virtually) closed (C) in the interspike interval, activate and abruptly open (O) and then quickly inactivate (I) during the action potential, and then recover from inactivation in the interspike interval. Interestingly, as it recovers from inactivation, the channel bypasses the open state, instead following the transition pathway indicated by the blue arrow in the figure. The resulting hysteresis (red vs. blue arrows) serves a fundamental role, as it effectively separates the process of inactivation during the action potential, controlled by the O - I transition, from the recovery from inactivation during the interspike interval, separately controlled by the I - C transitions.

As a result of separating these pathways, inactivation from the open state proceeds very quickly, giving the channel just enough time to flow current and sufficiently depolarize the membrane and activate other voltage-gated ion channels. In contrast, recovery out of inactivation proceeds more slowly from the closed states, at a rate that determines a refractory period compatible with the maximal spiking rate of the neuron. Furthermore, bypassing the open state minimizes the flow of $Na^+$ ions in the wake of the action potential and thus economizes the energy utilized by cellular ionic pumps (*Carter and Bean, 2009*). A conceptual kinetic mechanism that adequately captures all these properties is shown in *Figure 1D* (*Kuo and Bean, 1994*).

### Nav channels have long-term inactivation

Fast voltage-dependent activation (sub-millisecond) and inactivation (millisecond), as well as relatively fast recovery from inactivation (milliseconds), are kinetic properties common to all Nav channel subtypes, as described since the pioneering work of Hodgkin and Huxley (*Hodgkin and Huxley, 1952*; *Armstrong and Bezanilla, 1974*; *Armstrong and Bezanilla, 1977*; *Bezanilla and Armstrong, 1977*; *Aldrich et al., 1983*; *Vandenberg and Bezanilla, 1991a*; *Vandenberg and Bezanilla, 1991b*). However, the Nav kinetic inventory is richer than that, including such behavior as 'persistence' (*French et al., 1990*; *Crill, 1996*) or 'resurgence' (*Raman and Bean, 1997*; *Raman and Bean, 2001*). In raphe and other neurons, Nav channels exhibit yet another interesting property: when subjected to brief, repetitive depolarizations that mimic trains of action potentials, the $Na^+$ current

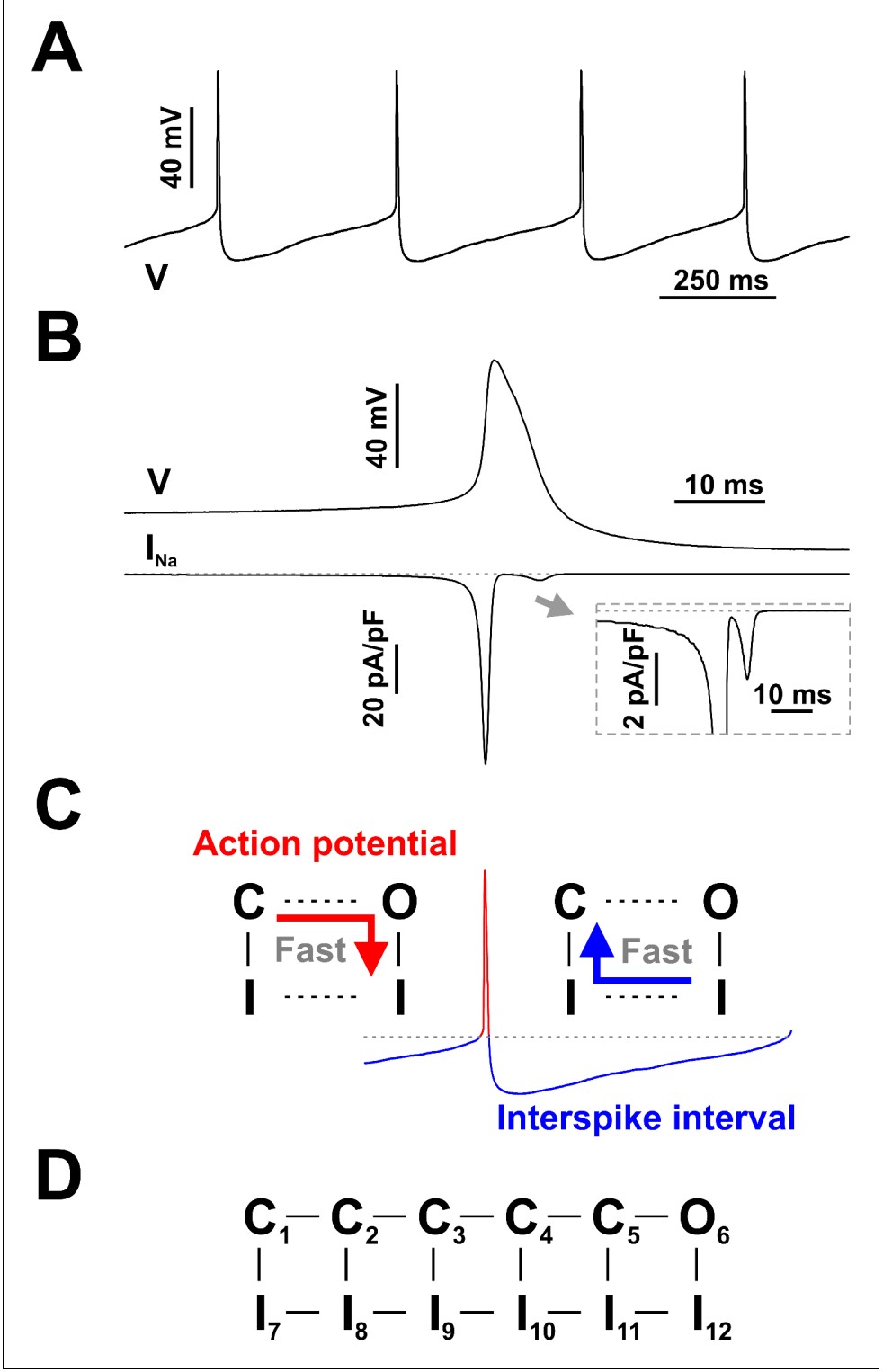

**Figure 1.** Spontaneous firing in serotonergic raphe neurons and the contribution of Nav channels. (**A** and **B**) Raphe neurons are characterized by slow and regular spiking and broad action potentials, with the spike-generating sodium current ($I_{Na}$) mostly restricted to the depolarization phase. (**C**) Schematic of state transitions undertaken by Nav channels during the spiking cycle (C - closed, O - open, I - inactivated states). (**D**) Conceptual Nav state model proposed to explain the fundamental kinetic properties of $I_{Na}$ in mammalian central neurons (**Kuo and Bean, 1994**). The representative current clamp recordings in (**A**) and (**B**) were obtained from RO neurons

*Figure 1 continued on next page*

*Figure 1 continued*

in neonatal rat brainstem slices. In (B) $I_{Na}$ was calculated in real-time and injected in the cell using dynamic clamp, as in *Milescu et al. (2010b)*.

evoked by each pulse progressively diminishes to levels inversely proportional to the pulse repetition rate (*Figure 2A*).

Furthermore, we can identify not one but two exponential components in the time course of recovery from inactivation (*Figure 2B*). Following a brief (5 ms) depolarizing pulse that completely inactivates the channels, approximately 80% of the initially available current recovers fast (milliseconds), with a voltage-dependent time constant, whereas the remaining 20% recovers slowly (hundreds of milliseconds to seconds), also with a voltage-dependent time constant. In brain slice recordings from neonatal RO neurons, the fast and slow components have the following time constants $\tau$ and relative amplitudes $a$ (mean $\pm$ SE): at $-80$ mV, $\tau_{fast}$ = 3.14 $\pm$ 0.126 ms, $a_{fast}$ = 0.786 $\pm$ 0.013, $\tau_{slow}$ = 612 $\pm$ 76 ms, and $a_{slow}$ = 0.212 $\pm$ 0.007 ($n$ = 18 cells); at $-100$ mV, $\tau_{fast}$ = 1.45 $\pm$ 0.07 ms, $a_{fast}$ = 0.775 $\pm$ 0.017, $\tau_{slow}$ = 209 $\pm$ 39 ms, and $a_{slow}$ = 0.199 $\pm$ 0.01 ($n$ = 6). The two components take similar values in acutely isolated mature dorsal raphe neurons: at $-80$ mV, $\tau_{fast}$ = 8.54 $\pm$ 0.59 ms, $a_{fast}$ = 0.81 $\pm$ 0.022, $\tau_{slow}$ = 517 $\pm$ 126 ms, and $a_{slow}$ = 0.187 $\pm$ 0.017 ($n$ = 6); at $-100$ mV, $\tau_{fast}$ = 5.25 $\pm$ 0.69 ms, $a_{fast}$ = 0.74 $\pm$ 0.042, $\tau_{slow}$ = 247 $\pm$ 84 ms, and $a_{slow}$ = 0.252 $\pm$ 0.033 (n = 5). The presence of the slow component in both intact and acutely dissociated neurons confirms the idea that the observed phenomenology is a genuine manifestation of the Nav kinetic mechanism, and not an artifact, such as caused by action potential back-propagation and poor space-clamp (*Milescu et al., 2010a*).

Similar Nav properties (i.e., adapting response to pulse trains and partial slow recovery from inactivation) have been observed in other neuronal types, such as hippocampal pyramidal neurons (*Mickus et al., 1999*). Appropriately, this phenomenology has been termed 'prolonged inactivation' (*Jung et al., 1997*) or 'long-term inactivation' (LTI) (*Dover et al., 2010*), to distinguish it from slow inactivation, which is a different process whereby Nav channels slowly (hundreds of milliseconds to seconds, or more) become unavailable when held at depolarizing potentials, and also slowly return to full availability at hyperpolarizing potentials (*Ruff et al., 1988*; *Fleidervish et al., 1996*). In contrast, LTI represents fast entry into a long-lived inactivated state, from which recovery is very slow. Interestingly, the LTI entry and exit time constants differ by three orders of magnitude: entry in milliseconds at depolarizing potentials and recovery in seconds during hyperpolarization (*Mickus et al., 1999*; *Milescu et al., 2010b*). As we discuss next, this fast-slow duality has important functional consequences.

## Mechanistic consequences of long-term inactivation

To investigate the role of LTI, we use here a previously developed model that explains well all the observed kinetic properties of Nav channels in RO neurons, including LTI (*Milescu et al., 2010b*). The Nav state model generated in that study was derived from a comprehensive collection of data that covered steady-state properties (activation and inactivation), as well as transient properties (time course of activation and inactivation, slow inactivation, recovery from inactivation, closed-state inactivation, entry into LTI, cumulative inactivation, etc.), all at multiple voltages and time scales (tens of microseconds to seconds). The rate constants were extracted from data using the optimization and parameter constraining algorithms implemented in the QuB software (*Navarro et al., 2018*; *Salari et al., 2018*). The model was then verified in live RO neurons, using the real-time computation algorithms implemented in QuB (*Milescu et al., 2008*), confirming that a model-generated current that replaced the TTX-blocked endogenous $I_{Na}$ was able to generate action potentials of similar shape and frequency. Interestingly, different studies in other neuronal types have arrived at conceptually similar models to explain LTI, although they were not necessarily as comprehensively constrained by experimental data (*Goldfarb et al., 2007*; *Menon et al., 2009*).

The model that explains LTI in RO neurons is shown in *Figure 3A*, where the basic kinetic scheme introduced in *Figure 1D* has been augmented with a non-conducting state ($S_{13}$) connected to the open state ($O_6$). The rate constants are explained in Materials and methods, and are as in *Milescu et al. (2010b)*. $S_{13}$ is a long-lived state representing the open channel blocked by the

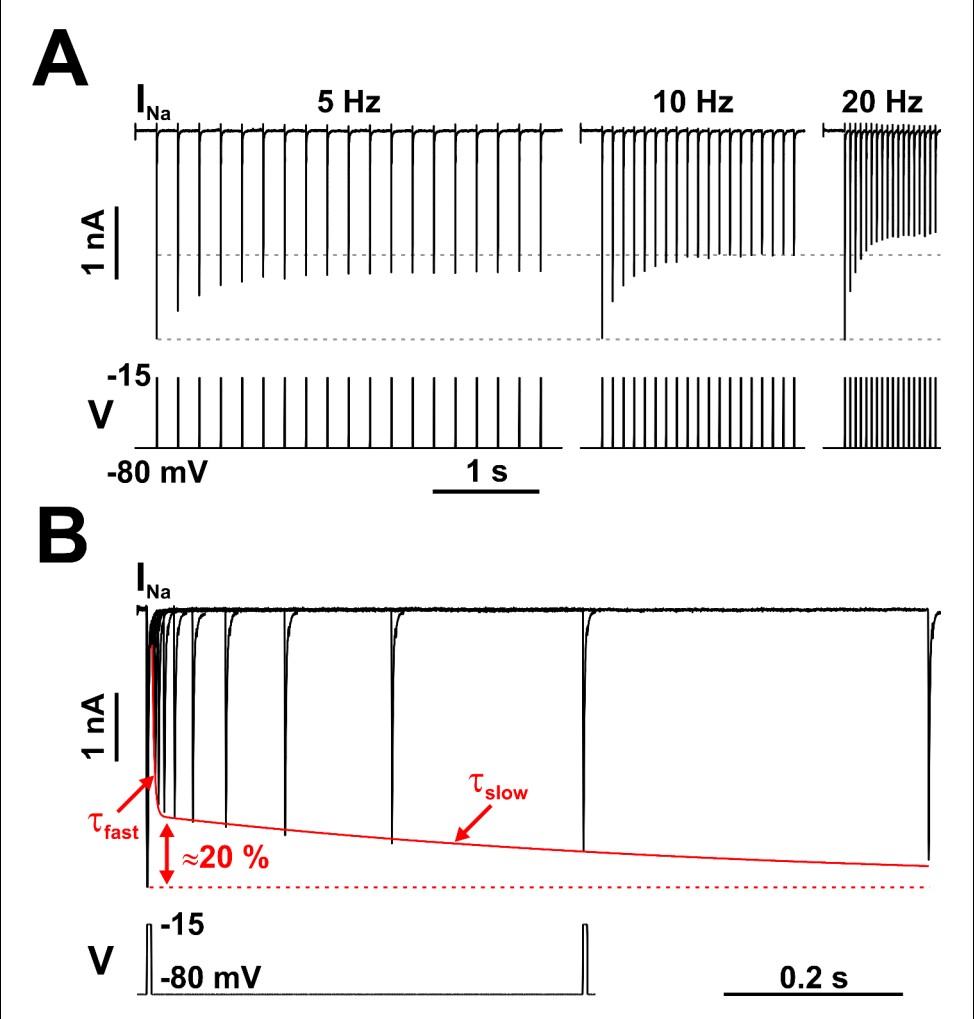

**Figure 2.** Nav channels in serotonergic raphe neurons exhibit a slow kinetic component. (**A**) The fraction of Nav channels available to generate current decays exponentially, when tested with trains of brief depolarizing voltage pulses (5 ms at −15 mV, repeated at 5, 10, or 20 Hz). The decay is greater at higher repetition rates. Each pulse completely inactivates $I_{Na}$, which then partially recovers from inactivation in the subsequent hyperpolarizing interval, at −80 mV. (**B**) The timing of recovery from inactivation was tested with a two-pulse protocol, where the first pulse (5 ms at −15 mV) inactivates the channels and the second pulse tests availability versus time, at −80 mV. As indicated by the two time constants ($\tau_{fast}$ and $\tau_{slow}$), recovery from inactivation is a bi-exponential process, with the slow component accounting for approximately 20% of the total current. The representative voltage clamp recordings in (**A**) and (**B**) were obtained from RO neurons in neonatal rat brainstem slices, and are TTX-subtracted. Statistical values are given in the main text.

The online version of this article includes the following source data for figure 2:

**Source data 1.** Recovery from inactivation of $Na^+$ current in rat neonatal RO neurons in brain slices and in acutely isolated mature dorsal raphe neurons, as shown in panel B.

putative auxiliary factor, FHF. The $O_6$ - $S_{13}$ rate constants take values that, in the context of the intrinsic Nav kinetics, result in the observed LTI phenomenology, that is, the complete inactivation induced by a brief depolarization and the fast-slow bi-exponential recovery at hyperpolarizing potentials. The $O_6$ - $S_{13}$ transition competes with the normal inactivation process that corresponds to the $O_6$ - $I_{12}$ transition, with interesting mechanistic consequences. A brief depolarizing pulse from −80 to 0 mV (*Figure 3B*) takes the channel rapidly through the sequence of closed states, as voltage sensors activate. Once it reaches the open state, the channel has now two distinct pathways to follow: to inactivate 'normally' into the $I_{12}$ state, or to long-term inactivate into the $S_{13}$ state. According to

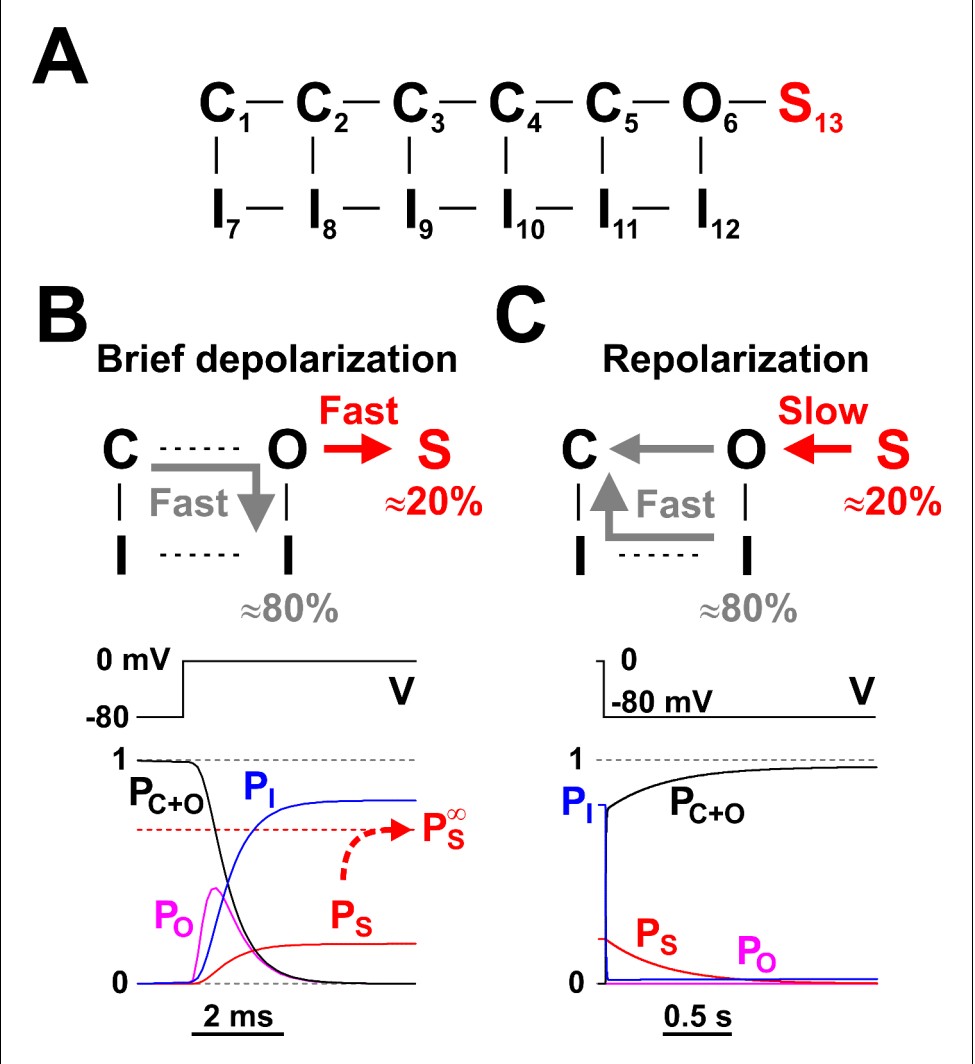

**Figure 3.** Nav long-term inactivation. (**A**) Conceptual Nav state model that adds one non-conducting state ($S_{13}$) to the model shown in *Figure 1D*, to explain the slow kinetic component illustrated in *Figure 2*. $S_{13}$ is a state of long-term inactivation. (**B** and **C**) State transitions undertaken by the channel during a brief depolarization (**B**) and in the subsequent hyperpolarizing interval (**C**); note the difference in time scales. During depolarization, ≈ 80% of channels follow the standard C - O - I pathway, whereas the remaining ≈ 20% enter the S state. The O - I and O -S transitions are both fast, as illustrated in the bottom left panel, and compete with each other. When the membrane potential returns to more negative values, the fraction of channels in the I states recovers quickly, whereas the S state fraction recovers slowly, explaining the bi-exponential recovery from inactivation illustrated in *Figure 2B*. $P_{C+O}$, $P_O$, $P_I$, and $P_S$ represent occupancies of closed plus open, open, inactivated, and long-term inactivated states, respectively.

The online version of this article includes the following source data for figure 3:

**Source data 1.** Rate constant values for the Nav kinetic model shown in panel A.

our experimental data (*Figure 2B*), ≈ 80% of the channels 'choose' the normal inactivation pathway and ≈ 20% take the LTI.

The $I_{12}$ and $S_{13}$ occupancy probabilities rise very quickly, reaching their respective values of 0.8 and 0.2 in a couple of milliseconds, as indicated by the time course of $P_I$ and $P_S$ in *Figure 3B* (lower panel). However, these are not the equilibrium values: if the channels are further maintained at depolarizing potentials for seconds, $P_S$ very slowly reaches a considerably higher value of ≈ 0.7, whereas $P_I$ drops accordingly to ≈ 0.3. Hence, entry into the $S_{13}$ state is a bi-exponential process, with one component fast enough to reach completion during a brief voltage pulse – or action

potential – and the other very slow, requiring several seconds to equilibrate. Nevertheless, since prolonged depolarizations are less likely to occur, the fast component is the more physiologically relevant one.

What happens after a brief depolarizing pulse, upon repolarization? As depicted in *Figure 3C*, those channels residing in the normal inactivated state $I_{12}$ cycle relatively fast (5 – 10 ms) back into the non-inactivated closed states (C), without visiting the open state. In contrast, channels residing in the LTI state $S_{13}$ recover slowly, in seconds, reaching the non-inactivated states through the open state. Therefore, the sum occupancy probability of all non-inactivated states (C states plus $O_6$ state; $P_{C+O}$ in *Figure 3C*) rises from 0 to $\approx$ 1, on a bi-exponential time course. The fast component corresponds to recovery from the normal inactivation process ($P_I$), whereas the slow component represents recovery from the LTI state ($P_S$). This dual fast-slow process explains the observed bi-exponential recovery from inactivation of $I_{Na}$ in serotonergic raphe neurons (*Figure 2B*). Interestingly, the open probability ($P_O$) remains close to zero, because recovery from the LTI state through the open state is stretched over a long time interval.

## The Nav channel as a molecular leaky integrator

We now arrive at our main idea, that Nav channels implement a molecular leaky integrator, through an interaction with auxiliary factors. Mathematically, a continuous-time leaky integrator is governed by the differential equation:

$$\frac{\mathrm{d}y(t)}{\mathrm{d}t} = x(t) - y(t)/\tau_{\mathrm{leak}}, \tag{1}$$

where $y$ is the output signal, $x$ is the input signal, $t$ is time, and $\tau_{\mathrm{leak}}$ is the leak time constant. Leaky integration can be more easily understood in the discrete time domain, as a mathematical operation that recursively calculates an output signal $y$ from an input signal $x$, as follows:

$$y_{t+\delta t} = y_t \times e^{-\delta t/\tau_{leak}} + x_{t+\delta t} \times \delta t, \tag{2}$$

where $\delta t$ is the sampling time between two measurements. When $\tau_{\mathrm{leak}}$ is infinite (or $\tau_{\mathrm{leak}} >> \delta t$), the above equation reduces to a simple integration (or summation), where $y_{t+\delta t} = y_t + x_{t+\delta t} \times \delta t$. If the input signal contains brief (sample-long) digital pulses (i.e., 0 or 1), the integrator becomes an event counter. When $\tau_{\mathrm{leak}}$ is zero (or $\tau_{\mathrm{leak}} << \delta t$), the output signal becomes a scaled copy of the input signal, where $y_{t+\delta t} = x_{t+\delta t} \times \delta t$. Otherwise, when $\tau_{\mathrm{leak}}$ takes a finite value, the output signal at a given time point is first 'leaked' (i.e., reduced) by a factor determined by the ratio $\delta t/\tau_{\mathrm{leak}}$, and then 'integrated' (i.e., added) with the input signal arriving at the next time point, to calculate the next output value.

How does leaky integration apply to Nav channels? First, we consider the membrane potential ($V_m$) to be the input signal, and the occupancy of the LTI state $S_{13}$ ($P_S$) to be the output signal. We further consider $V_m$ to be 'digital' and take only two states: 'low' in the interspike interval and 'high' during the action potential, with the 'low' state meaning functionally zero input (nothing to integrate). Finally, we consider that the time scale of the leak process is orders of magnitude longer than the width of an action potential. Under these conditions, the integration step corresponds to quickly incrementing $P_S$ whenever the input signal $V_m$ switches to a high state (an action potential), as shown in *Figure 4A*, and the leak step corresponds to slowly decrementing $P_S$ whenever $V_m$ switches to a low state (the interspike interval), as shown in *Figure 4B*.

As indicated by our experimental data (*Figure 2B*) and correctly predicted by our model, the arrival of a brief depolarization, such as an action potential, prompts $\approx$ 20% of all available (non-inactivated) channels to rush into the long-term inactivated state $S_{13}$, whereas the other $\approx$ 80% quickly occupy the inactivated state $I_{12}$, with time courses as shown in *Figure 3B*. Channels that are already inactivated will maintain their state. Because $P_S$ is in fact a probability bound by 1, it cannot increase indefinitely. Thus, each action potential can only increase $P_S$ by $\approx$ 20% of the fraction of available channels. At the beginning of a spike train, when all channels are available, this increment is $\approx$ 0.2 but progressively gets smaller, as fewer channels remain available (*Figure 4C*). The specific value of the increment depends on the relative kinetics of the $O_6$ - $S_{13}$ and $O_6$ - $I_{12}$ transitions. The bi-exponential time course of $P_S$, with a fast component that completes in $\approx$ 2 ms, coupled with a much slower component, ensures that changing the width of the action potential would not

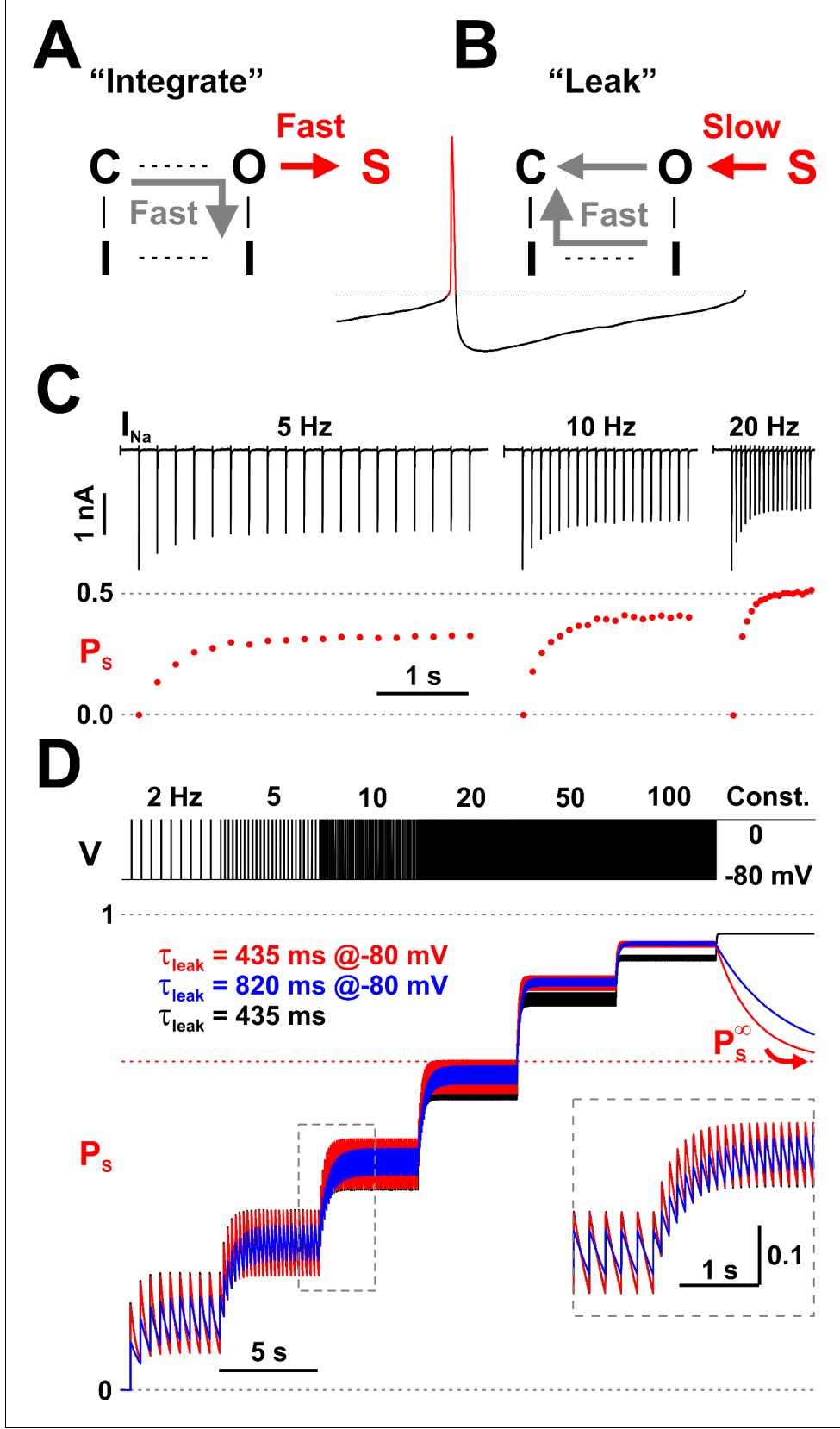

**Figure 4.** Nav channels implement a molecular leaky integrator that measures spiking frequency. (**A**) The 'integration' is represented by the quick entry of channels into the LTI state S, during an action potential. (**B**) The
*Figure 4 continued on next page*

*Figure 4 continued*

'leak' corresponds to the slow transition out of the S state, during the interspike interval. (**C**) The average occupancy of the LTI state ($P_S$) increases with pulse repetition rate. The current ($I_{Na}$) trace is as in **Figure 2A**. (**D**) Testing the leaky integrator with trains of brief voltage pulses (5 ms at 0 mV, from −80 mV), with different repetition rates (2 to 100 Hz) or at constant depolarization. The average occupancy of the S state is a function of pulse frequency. The Nav model in **Figure 3A** was tested with two sets of kinetic parameters for the $O_6$ - $S_{13}$ transition, corresponding to $\tau_{leak}$ = 435 ms and 80%/20% normal inactivation vs. LTI ratio (red trace; $k_{6,13}$ = $400.8 \times e^{-0.011 \times V}$ and $k_{13,6} = 0.207 \times e^{-0.031 \times V}$) or 820 ms and 90%/10% ratio (blue trace; $k_{6,13} = 205.7 \times e^{-0.011 \times V}$ and $k_{13,6} = 0.106 \times e^{-0.031 \times V}$). Both sets have $P_S^{\infty} \approx 0.7$. For comparison, the response of a discrete-time mathematical leaky integrator with $\tau_{leak}$ = 435 ms (black trace; **Equation 3**).

significantly alter the 80%/20% ratio, unless the width were shorter than 2 ms, in which case LTI would be reduced, or if the depolarization were extended to hundreds of milliseconds or seconds, in which case $P_S$ would eventually reach an equilibrium value of ≈ 0.7, as represented by $P_S^{\infty}$ in **Figure 3B**.

Once the action potential ends and $V_m$ switches to the 'low' state, $P_S$ starts to 'leak', because the input signal is 'zero' in the interspike interval and there is nothing to 'integrate'. Thus, $P_S$ decays exponentially, with the same time constant as the slow component of recovery from inactivation (hundreds of milliseconds). Those channels that were long-term inactivated (occupying the $S_{13}$ state) at the beginning of the interspike interval will remain unavailable to conduct current for an accordingly long time interval, on the order of seconds. In contrast, those channels that were normally inactivated (occupying the $I_{12}$ state) will become available much sooner, after only a few milliseconds. If another action potential were triggered while some channels were still residing in either the long-term inactivated state $S_{13}$ or in the inactivated state $I_{12}$, the available (non-inactivated) channels will again divide 80%/20% between normal inactivation and long-term inactivation, and so on. As a result, the occupancy of the $S_{13}$ state would keep increasing with each action potential by progressively smaller increments, unless the interspike intervals were long enough to allow complete recovery out of the $S_{13}$ state.

## Nav channels detect action potentials and measure spiking frequency

The use of an integrator is obvious – to summate, to count – but what about a 'leaky' integrator? As it happens, a leaky integrator not only describes many real-world phenomena, such as rain accumulating into a lake that drains into a river, but also has many technical applications. One of the most obvious is to convert the frequency of an input signal into the amplitude of an output signal. For example, in our own electrophysiology backyard, it is customary to pass a nerve signal through a leaky integrator (implemented in hardware or software) to convert noisy and dense spike trains into an amplitude signal that more legibly indicates the frequency of those spikes.

As a molecular leaky integrator, Nav channels can also 'measure' the frequency of action potentials and 'store' it in the occupancy of the LTI state $S_{13}$. As demonstrated by the experimental data shown in **Figure 2A**, the amount of Na$^+$ current evoked with 5 ms depolarizing pulse trains decays exponentially with each pulse, and decays more at higher repetition rates. In **Figure 4C**, we calculate and plot $P_S$ using the equation $P_S = 1 − I_{Na}^P / I_{Na}^0$, where $I_{Na}^P$ is the peak Na$^+$ current raised by pulse $p$ and $I_{Na}^0$ is the peak Na$^+$ current raised by the first pulse in the series. The rationale of using this equation to obtain $P_S$ is that the inter-pulse interval (195, 95, and 45 ms in this case) is long enough to allow full recovery from the normal inactivation, whereas recovery from the LTI state will be incomplete, as we know from the data shown in **Figure 2B**. Thus, having a fraction of channels still trapped in the LTI state will proportionally reduce the maximal $I_{Na}$. As seen for this representative data set, $P_S$ reaches steady values of ≈ 0.3 at 5 Hz, ≈ 0.4 at 10 Hz, and ≈ 0.5 at 20 Hz, exhibiting a nonlinear dependence on the pulse repetition rate.

To better understand how Nav channels may interpret action potential frequency, we simulated the response of our Nav model to a train of 5 ms depolarizing pulses from −80 to 0 mV, repeated at different rates, and compared it with the response of a mathematical leaky integrator. For the sake of simplicity, the mathematical leaky integrator was presented with the same train of pulses as the Nav model, but the input variable $x$ was assigned 0 and 1 values, instead of −80 and 0, respectively. To match the behavior of the Nav model, we also modified **Equation 2**, as follows:

$$y_{t+\delta t} = y_t \times e^{-\delta t/\tau_{leak}} + x_{t+\delta t} \times f_{t+\delta t}, \tag{3}$$

$$f_{t+\delta t} = y_{\text{inc}} \times (\delta t/t_p) \times (1 - y_t \times e^{-\delta t/\tau_{leak}}), \tag{4}$$

where $y_{\text{inc}} = 0.2$ represents the 20% maximum increase in the output variable $y$, spread over the duration $t_P$ of a pulse. Thus, $f$ is an ad hoc expression that ensures that, during a pulse in the input variable $x$, the output variable $y$ can only increase by 20% of the difference between one and its current value (i.e., 20% of $1 - y_t \times e^{-\delta t/\tau_{leak}}$), in the same way as $P_S$ can only increase by 20% of the fraction of available channels (i.e., 20% of $1 - P_S$). As a result, the range of the output variable $y$ is restricted between 0 and 1, similarly to $P_S$.

Our Nav model has a leak time constant $\tau_{leak} \approx 435$ ms at $-80$ mV, a normal inactivation/LTI ratio of 80%/20%, and an equilibrium S state occupancy $P_S^\infty \approx 0.7$. These quantities depend on the specific values and voltage dependence of the $O_6$ - $S_{13}$ rate constants ($k_{6,13} = 400.8 \times e^{-0.011 \times V}$ and $k_{13,6} = 0.207 \times e^{-0.031 \times V}$), in the context of all the other rate constants. Keeping the exponential factors $k_{6,13}^1$ and $k_{13,6}^1$ and all other rate constants unchanged, $\tau_{leak}$ depends mostly on the pre-exponential factor $k_{13,6}^0$ (lower value increases $\tau_{leak}$), the normal inactivation/LTI ratio depends on the pre-exponential factor $k_{6,13}^0$ (lower value increases the ratio), and $P_S^\infty$ depends on the ratio between $k_{6,13}^0$ and $k_{13,6}^0$ (lower ratio decreases $P_S^\infty$).

As indicated by the red trace in *Figure 4D*, when the Nav model is presented with a train of depolarizing pulses, $P_S$ reaches levels that depend nonlinearly on the pulse repetition rate. At any given pulse frequency, $P_S$ oscillates between a maximum reached at the end of a pulse, and a minimum reached at the end of the inter-pulse interval. At the maximum tested frequency of 100 Hz, $P_S$ oscillates minimally between 0.92 and 0.93. Then, under constant stimulus, $P_S$ decays down to its equilibrium value $P_S^\infty$ (see *Figure 3B*). For comparison, we modified the $O_6$ - $S_{13}$ rate constants ($k_{6,13} = 205.7 \times e^{-0.011 \times V}$ and $k_{13,6} = 0.106 \times e^{-0.031 \times V}$) to obtain $\tau_{leak} \approx 820$ ms and 90%/10% normal inactivation/LTI ratio, while keeping the same $P_S^\infty$. As indicated by the blue trace in *Figure 4D*, in this case $P_S$ exhibits reduced oscillations and takes longer to reach steady-state at a given pulse rate but follows the same overall trend with the increase in stimulus frequency. The discrete-time mathematical leaky integrator responds in a similar fashion (black trace, *Figure 4D*), although it starts to deviate at higher frequencies and particularly under continuous input, where it reaches a maximum, whereas the Nav model shows $P_s$ slowly decaying to $P_S^\infty$.

## Nav channels drive spiking frequency

As demonstrated in *Figure 4C and D*, Nav channels respond to stimulation frequency by changing the occupancy of the long-term inactivated state, which, in turn, changes the amount of available $I_{Na}$. This observation raises the reciprocal question: does a neuron respond to the amount of available $I_{Na}$ by changing its spiking frequency? In other words, if Nav channels 'encode' frequency via $S_{13}$, can they also 'decode' it via $I_{Na}$? $I_{Na}$ can potentially drive spiking frequency via two interrelated mechanisms: directly, by controlling the rate of depolarization in the interspike interval, and indirectly, by shaping the action potential waveform and thus affecting the other ionic currents that flow during the action potential depolarization and during the ensuing interspike interval. To test these possibilities, we used dynamic clamp (*Sharp et al., 1993*) to inject a model-based $I_{Na}$ in RO neurons, and measured the response of the cell to increasing levels of Nav conductance ($G_{Na}$). We tested both non-LTI (*Figure 1D*) and LTI (*Figure 3A*) models. The endogenous $Na^+$ current was blocked with TTX.

Indeed, RO neurons do respond to the level of available $I_{Na}$ by changing their spiking frequency, as shown in *Figure 5A*. The response in frequency vs. $G_{Na}$ is approximately linear over the tested range of $G_{Na}$ (5 to 20 nS/pF; *Figure 5B*), regardless of which Nav model generates $I_{Na}$ (LTI, red symbols and fit line, or non-LTI, black symbols and fit line). However, for the same value of $G_{Na}$, the LTI model results in lower spiking frequency than the non-LTI model. This difference is explained by the reduced amount of dynamically available $I_{Na}$ generated by the LTI model.

The effect of $I_{Na}$ on firing is further examined in *Figure 5C*, where three example waveforms, each containing two action potentials and the interspike interval in between, obtained with the LTI and non-LTI models under different $G_{Na}$ values, are shown aligned to the peak of the first action

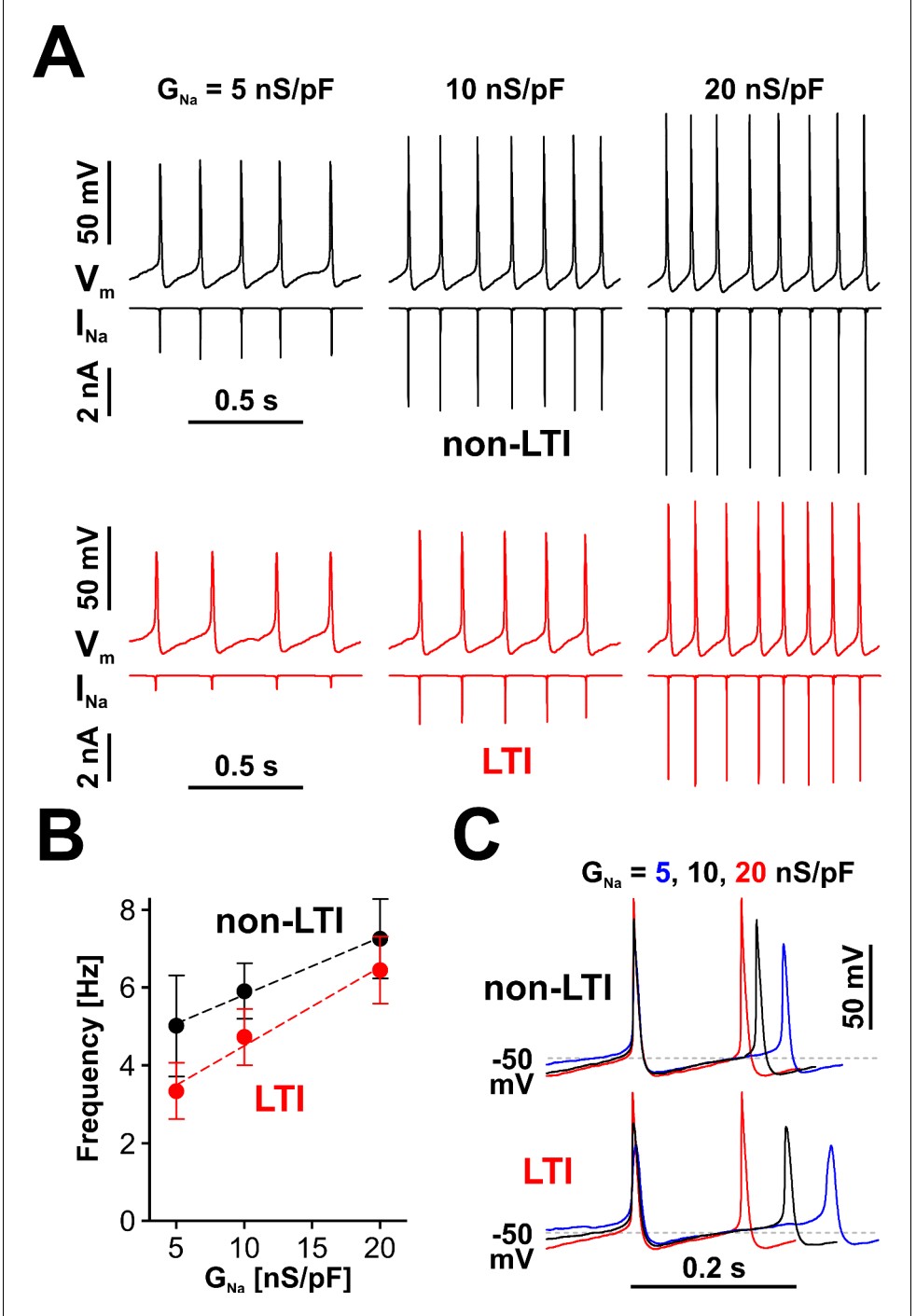

**Figure 5.** Nav channels drive spiking frequency in serotonergic raphe neurons. (**A**) Representative dynamic clamp recordings, where $I_{Na}$ generated by a non-LTI (black traces) or LTI (red traces) Nav model was injected in a neuron, under different levels of Nav conductance ($G_{Na}$). The endogenous sodium current was blocked with bath-applied TTX. (**B**) Spiking frequency increases proportionally with $G_{Na}$, over a range typical for neonatal RO neurons, with $I_{Na}$ generated by either the non-LTI model (black symbols and fit line, mean ± SE; slope = 0.148 ± 0.011, intercept = 4.34 ± 0.14, n = 7, F-test, p=0.13) or the LTI model (red symbols and fit line; slope = 0.202 ± 0.026, intercept = 2.48 ± 0.346, n = 7, F-test, p=0.0093). The two datasets are statistically different (paired two-tailed t-test, p=0.0394). (**C**) Representative dynamic clamp traces obtained with $I_{Na}$ generated by the non-LTI (top) or LTI (bottom) model, illustrating how the action potential and the interspike interval are shaped by $G_{Na}$. The data were obtained without stimulation ($I_{Inj}$ = 0) from RO neurons in neonatal rat brainstem slices.

*Figure 5 continued on next page*

*Figure 5 continued*

The online version of this article includes the following source data for figure 5:

**Source data 1.** Frequency *vs.* $G_{Na}$ for LTI and non-LTI models, as shown in panel B.

potential. As can be observed, the rate of membrane depolarization in the interspike interval is relatively independent of $I_{Na}$ and spiking frequency, from the hyperpolarization occurring immediately after an action potential, up to $\approx -50$ mV. From this point, the rate of depolarization changes proportionally to $G_{Na}$ and determines the duration of the remaining interspike interval. As a result, the firing frequency will eventually saturate with increasing amounts of available $I_{Na}$, which explains why the LTI and non-LTI models appear to converge to similar spiking frequencies, as $G_{Na}$ is increased, and why the non-LTI model shows a reduced frequency vs. current slope, compared to the LTI model (*Figure 5B*). Another interesting point is that at low firing frequency Nav channels spend more time at membrane potentials where closed-state inactivation occurs (between $-75$ and $-50$ mV), which explains why $I_{Na}$ for the LTI model is disproportionately smaller than at higher frequencies (*Figure 5A*, the red $I_{Na}$ trace). With both models, the amount of dynamically available $I_{Na}$ changes the overall action potential shape (*Figure 5C*), particularly the peak value, presumably changing other currents and thus indirectly affecting spiking frequency.

## Firing rate is lower in the presence of LTI

The experiments described in *Figures 2* and *5* clearly demonstrate that not only can Nav channels sense the frequency of action potentials, but they also can drive the firing rate in RO neurons, with the LTI model resulting in lower frequency. What about firing under excitatory input? Considering these two reciprocal relationships, one can predict that changes in excitatory input will have a reduced effect on firing frequency in neurons that express Nav channels with LTI. The reason is that an increase in excitation will initially determine an increase in frequency, but this elevated frequency will then push more Nav channels into the LTI state, thus reducing the available $I_{Na}$ and returning frequency to lower values.

We tested this prediction by subjecting neurons to increasing levels of depolarizing bias current ($I_{Inj}$), with the endogenous Na⁺ current TTX-blocked and replaced via dynamic clamp with $I_{Na}$ generated by either the LTI or the non-LTI model. We knew from our previous work (*Milescu et al., 2010b*) that RO neurons express a strong firing adaptation mechanism, unrelated to sodium channels, and the challenge was to separate the effects of LTI from this other mechanism. Another challenge was maintaining stability throughout a long recording protocol, given how sensitive firing frequency is to small fluctuations in patch properties in these neurons. As a compromise solution, we designed the dynamic clamp protocols shown in *Figure 6A and B*, with the only difference between them being the order in which the two Nav models are alternated. However, we also tried protocols where $I_{Inj}$ was stepped through different values for each model separately, with each depolarization step preceded by a five-second rest period at $-65$ mV, and obtained similar results.

As illustrated in *Figure 6C*, where we extract the instantaneous spiking frequency from the example traces shown in *Figure 6A and B*, RO neurons respond immediately with an increase in their firing rate when $I_{Inj}$ is stepped up. However, this initial increase in frequency is followed by a slow decay, which can be quite dramatic at greater depolarization (i.e., when $I_{Inj} = 40$ pA). Interestingly, this decay occurs when either model is active, but it is deeper with the LTI model. During a given depolarization episode, the frequency increases when changing the active model from LTI to non-LTI and decreases with the opposite change. According to *Figure 6D*, where we plot the stationary spiking frequency versus the level of depolarization, the neuron generally maintains a lower firing rate when Nav channels have LTI, and the relative difference between LTI and non-LTI models is larger at greater depolarizations. Regardless of the model, the frequency slightly increases with $G_{Na}$, in line with the results shown in *Figure 5B*. Altogether, these results make it clear that a neuron with the LTI model has a significantly flatter stationary response to depolarizing input, but it remains equally capable of high-frequency transients. Furthermore, these data emphasize how strongly RO neurons adapt to depolarizing input, via a combination of mechanisms, of which only one is based on Nav LTI. The results of swapping LTI and non-LTI models in the same cell (*Figure 6C*, red vs. black traces) are to some extent masked by these other adaptation mechanisms.

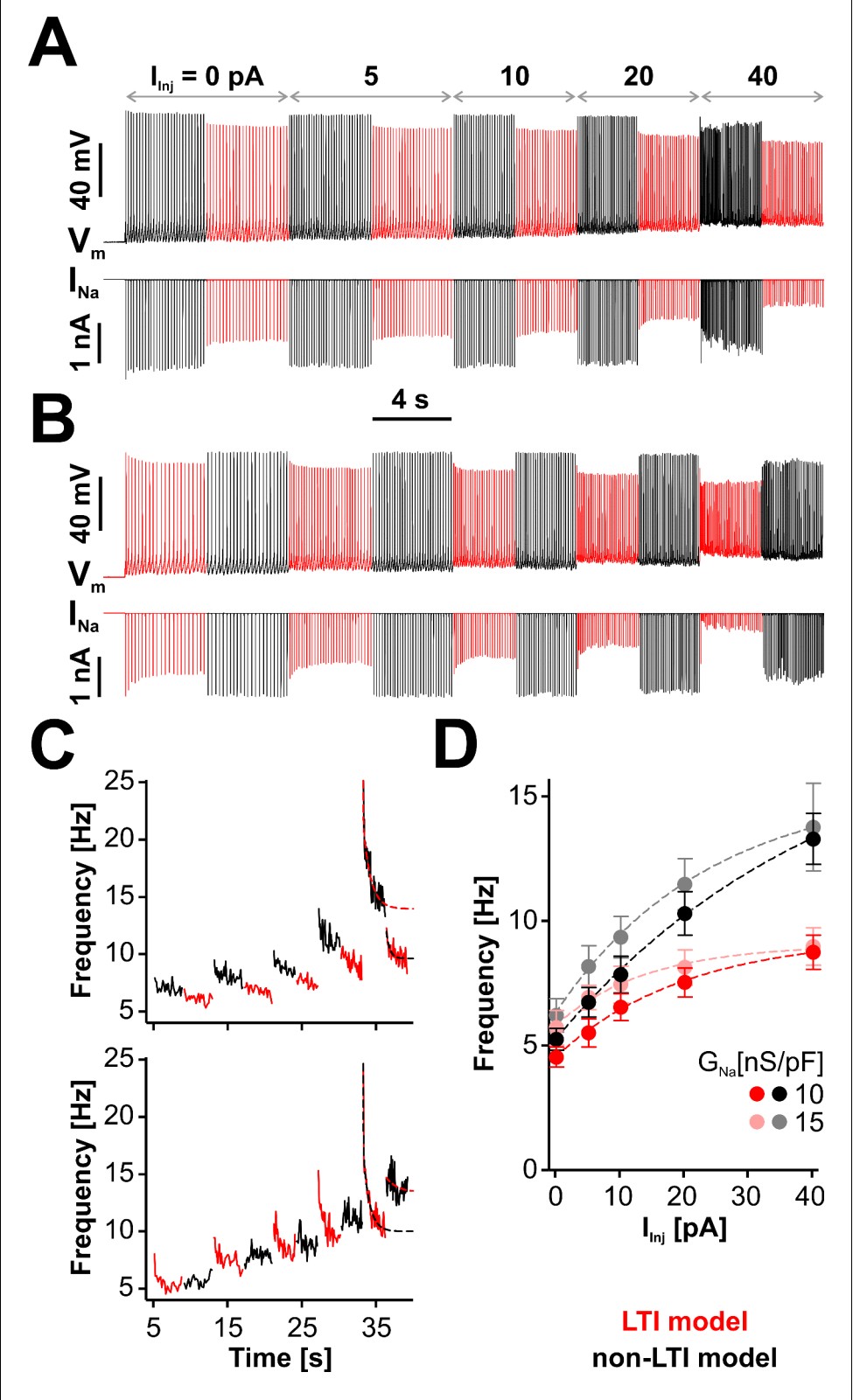

**Figure 6.** LTI helps the neuron to maintain a low spiking frequency against sustained depolarizations. (A) Representative dynamic clamp recording, where $I_{Na}$ (lower trace) generated by the LTI (red) or non-LTI (black) model was injected in a neuron, under increasing depolarizing current ($I_{Inj}$). Both models are integrated throughout the entire protocol, but only one model at a time injects current in the cell. In this example, $G_{Na}$ was

*Figure 6 continued on next page*

*Figure 6 continued*

10 nS/pF. Before the protocol started, the neuron was clamped at −65 mV for 5 s. (B) Same as in (A), but the models were alternated in the opposite order. (C) Instantaneous spiking frequency extracted from the recording shown in (A) (upper panel) or (B) (lower panel). (D) Quasi steady-state spiking frequency vs. $I_{Inj}$ (mean ± SE, n = 13). The values were obtained by averaging over the second half of each step, from traces obtained with the protocol shown in (A). The datasets are statistically different (paired two-tailed t-test) for LTI vs. non-LTI model (p=0.0387 or 0.0374, for $G_{Na}$ = 10 or 15 nS/pF, respectively) and for $G_{Na}$ = 10 vs. 15 nS/pF (p=0.0143 or 0.0038, for LTI or non-LTI model, respectively). The dashed lines in (C) and (D) are exponential fits meant as a visual aid. The data were obtained from RO neurons in neonatal rat brainstem slices.

The online version of this article includes the following source data for figure 6:

**Source data 1.** Frequency *vs.* $I_{Inj}$ for LTI and non-LTI models, as shown in panel D.

## Nav channels are a molecular controller that regulates neuronal firing rate

The overall idea that emerges from our experiments is that Nav channels implement a negative feedback loop that can regulate the firing frequency of the host neuron, as shown in *Figure 7A*. This negative feedback is conceptually similar to the typical process controller described in engineering applications (*Figure 7B*). As a technological example, one could take the home furnace as the 'process', the indoor temperature as the 'process variable', and human comfort as the 'product'. A sensor measures the temperature and forwards the measurement to a thermostat ('controller'), which calculates the difference ('error') between a user-prescribed value ('set point') and the measured value. A control algorithm processes the error and determines the timing and the amount of gas to burn in the furnace ('control variable'), such as to maintain the actual indoor temperature relatively constant and approximately equal to the desired temperature. The furnace will burn gas when the error is positive, which will increase the temperature and eventually make the error zero or negative, which in turn will instruct the furnace to stop burning gas, until the temperature drops again and the error becomes again positive. The thermostat can compensate for changes in the outside temperature (the 'disturbance').

The similarity between the Nav-based neuronal controller and the general process controller is quite striking. In the neuronal case (*Figure 7A*), we can identify the 'process' as the neuron, the 'process variable' as the spiking frequency, and the 'product' as serotonin. The Nav kinetic mechanism embodies the 'sensor', the 'set point', and the 'controller' altogether, as follows: i) spiking frequency is measured by the occupancy of the LTI state $S_{13}$, ii) the set point is determined by the rate constants of the $O_6$ - $S_{13}$ transition, and iii) the control algorithm is simply represented by the mutually exclusive relationship between the occupancy of the LTI state and the fraction of Nav channels available to generate current, which is equal to 1 - $P_S$. Unlike the general controller, which in principle can implement arbitrarily complex error correction (the P, I, and D blocks in *Figure 7B*), the Nav-based neuronal controller is limited by its simple physical implementation. The 'control variable' is represented by the amount of available $I_{Na}$, which can drive spiking frequency and thus closes the control loop. Finally, the 'disturbance' is represented by changes in synaptic or other stimuli ('input').

## Discussion

Nav channels play a fundamental role in cellular excitability, acting as a nonlinear amplifier that converts a small membrane depolarization into an action potential. Their intrinsically complex kinetic mechanism (*Armstrong and Gilly, 1979*; *Armstrong, 2006*) is further tweaked in different neuronal populations by interaction with auxiliary subunits and factors to create new functional behaviors (*Grieco et al., 2005*; *Aman et al., 2009*; *Ben-Johny et al., 2014*) that make Nav channels drive spontaneous spiking (*Do and Bean, 2003*), enable fast spiking (*Raman and Bean, 2001*; *Khaliq et al., 2003*), or establish complex firing modes (*Magistretti et al., 2006*; *Yamanishi et al., 2018*). We showed here that adding a state of long-term inactivation (*Figure 3*), as can be created by an interaction with auxiliary factors identified as FHFs (*Smallwood et al., 1996*; *Liu et al., 2003*; *Wittmack et al., 2004*; *Lou et al., 2005*; *Rush et al., 2006*; *Goldfarb et al., 2007*; *Dover et al., 2010*), significantly expands the Nav computational repertoire and creates new functional roles.

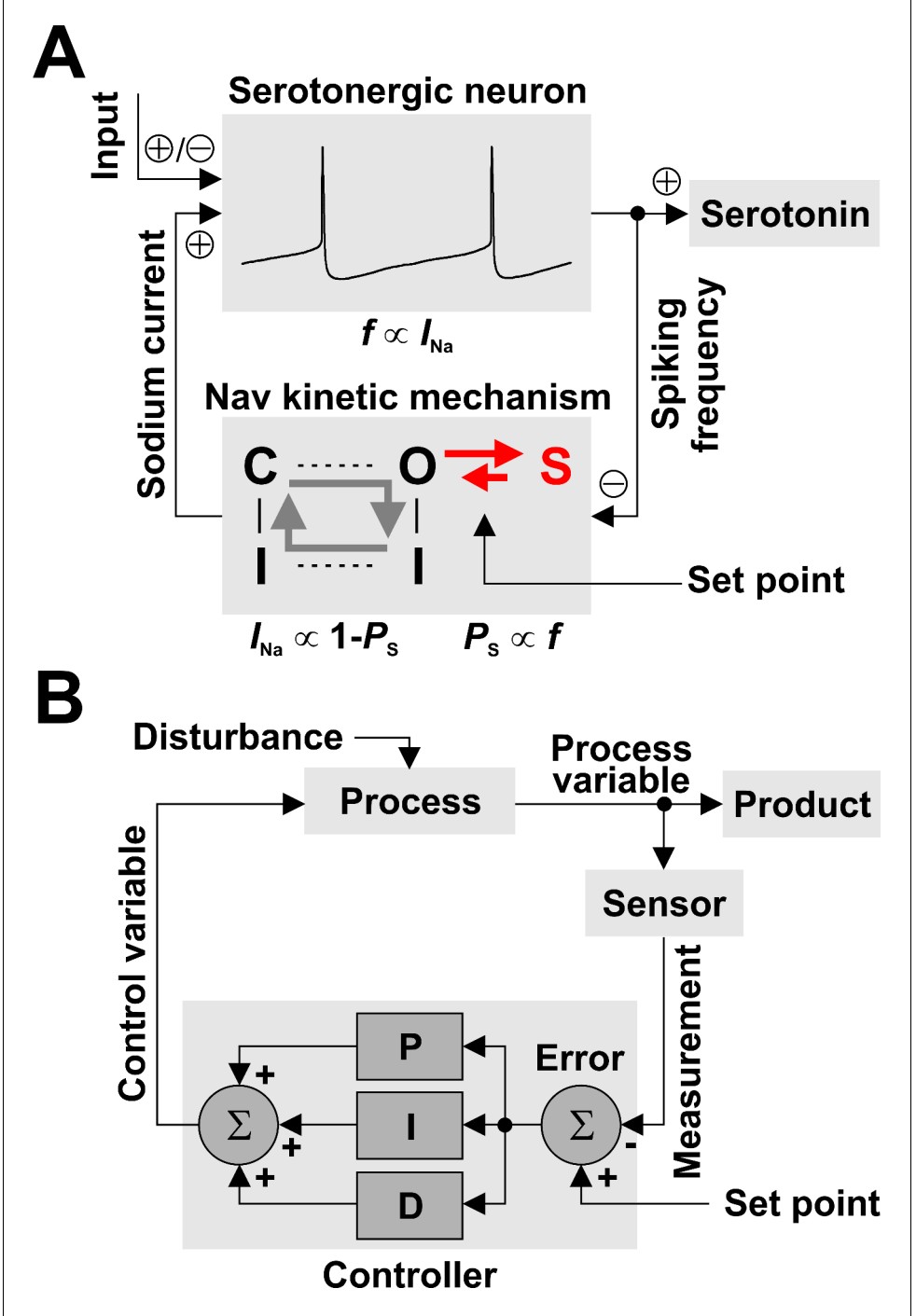

**Figure 7.** Nav channels can regulate spiking frequency in serotonergic raphe neurons in a negative feedback loop. (**A**) Nav-based neuronal controller, where spiking frequency $f$ is both 'measured' by the occupancy of the long-term inactivated state ($P_S$) and 'driven' by the amount of available $I_{Na}$. For example, an increase in $f$ via excitatory synaptic input causes an increase in $P_S$, which determines a decrease in the fraction of Nav channels available to generate current, equal to $1 - P_S$. In turn, this decrease in $I_{Na}$ reduces firing frequency, closing the loop. The kinetics of LTI establish the operating point of the control loop. (**B**) For comparison, a conceptual schematic of controllers used in engineering applications, which can combine multiple methods for error correction: proportional (P), integral (I), and derivative (D).

Our main finding is that LTI effectively turns the Nav channel into a molecular leaky integrator that can analyze the firing activity of the host neuron and encode its spiking frequency into the fraction of available Na$^+$ current (*Figure 4*). At the same time, Nav channels can drive the neuron to spike, with a frequency that depends on how much Na$^+$ current is dynamically available, which is modulated by LTI (*Figure 5*). These two reciprocal relationships between frequency and available current establish a negative feedback control loop that can regulate the frequency of action potentials (*Figure 7*). As a result, neurons expressing Nav channels with LTI would be less sensitive to changes in excitatory input and would generally maintain a lower firing rate (*Figure 6*).

Future work must explain why this LTI-based functionality is necessary, in general and specifically in the case of raphe neurons. Thanks in part to LTI, raphe neurons strongly adapt their action potential shape and frequency in response to stimuli (*Milescu et al., 2010b*; *Venkatesan et al., 2014*), but the role of this adaptation remains to be fully explained in the context of a more complete circuit that also includes the target neurons and all inputs, including auto-feedback. Raphe neurons act as a sprinkler-type system that provides serotonergic modulation to most brain regions (*Daubert and Condron, 2010*), via volume transmission carried by relatively evenly distributed innervation (*Azmitia and Whitaker-Azmitia, 1995*; *Aghajanian and Sanders-Bush, 2002*; *Vizi et al., 2010*). We can speculate that a downstream effect of a frequency control mechanism would be to release serotonin at more stable levels. In the long term, these levels could be dialed up and down by acting on multiple regulatory mechanisms, including LTI. In parallel with this increased stability conferred by LTI, raphe neurons would still maintain their capability of responding to short-term changes in stimuli, as shown in *Figure 6C*. For example, the *raphe obscurus* neurons investigated in this study are known to have mutual interactions with respiratory pacemaker neurons in the pre-Bötzinger Complex and briefly increase their firing rate in phase with respiratory network activity (*Ptak et al., 2009*).

It is interesting to note that LTI is not the only way for Nav channels to create feedback within the neuron. The most important is the well-known positive feedback mechanism that makes Nav channels the generator of action potentials. In the case of RO neurons, any depolarization that takes the cell above $\approx -50$ mV starts to activate Nav channels, which generate current that further depolarizes the cell, which in turn activates more channels, and so on, until an action potential is fired (*Figure 5C*). Yet another positive feedback may be created by closed-state inactivation. The interspike interval in RO neurons takes the characteristic shape of a steady ramp (see *Figures 1A* and *5C*) that is lengthened or shortened at lower or higher spiking frequencies, respectively. The ramp starts around $-70$ to $-60$ mV and ends at $\approx -45$ mV, whereas the Nav channel steady-state inactivation begins at $\approx -80$ mV and is almost complete at $-40$ mV. Longer interspike intervals would give Nav channels more time to equilibrate via closed-state inactivation, which would further slow down the ramp, because there would be less Na$^+$ current to drive the rate of depolarization and perhaps even insufficient current to generate an action potential. To escape this predicament, slow-spiking raphe neurons express Nav channels with a relatively low rate of closed-state inactivation (*Milescu et al., 2010b*), and also feature voltage-activated calcium channels that alone can sustain spontaneous spiking with very shallow action potentials and thus can act as a backup (unpublished data). In principle, LTI is a form of open-state block and should not occur from closed states, but the existence of closed-state LTI remains to be further investigated. Although these two forms of positive feedback (voltage-dependent activation and closed-state inactivation) are critical, they are confined to one action potential cycle, whereas LTI is a form of negative feedback that extends over many action potentials and thus can effectively regulate the firing rate.

Mathematically, LTI can be described as a leaky integrator that summates over time an input signal into an output signal, while the output steadily decays (*Equation 2*). Studying for an exam is a good analogy: we quickly accumulate knowledge with every page we read, while at the same time we slowly forget whatever we learned. In the field of electrical engineering, a simple leaky integrator can be implemented with an RC circuit. In the molecular world, nature has found a simple and elegant way of converting the Nav channel into a leaky integrator, via LTI, while preserving its basic functionality as a spike generator – and also partial pacemaking driver in RO neurons. LTI is a mechanism that renders a fraction of the total number of Nav channels functionally non-available, in the sense that these channels are trapped in a non-conducting state and thus cannot contribute current to an action potential. This fraction increases and decreases with dynamics dictated by the frequency of action potentials, effectively becoming a measure of neuronal activity. For this mechanism to

work, a single action potential must quickly increment the non-available fraction, whereas the subsequent interspike interval must slowly decrement it. In the case of LTI in RO neurons, the time constants of these two processes are separated by three orders of magnitude (*Figure 3B and C*).

The simplest kinetic mechanism compatible with the observed LTI behavior is shown in *Figure 3A*, where an additional non-conducting state is connected to the open state of the basic Nav mechanism shown in *Figure 1D* (*Dover et al., 2010*; *Milescu et al., 2010b*). This model predicts the critically important large discrepancy between LTI entry and exit time constants yet realistically assumes only minimal electrical charge for the LTI transition itself. Nevertheless, the LTI mechanism – as well as the intrinsic Nav mechanism itself – may be different and possibly more complex in reality, and needs further investigation. More work is also necessary to identify the specific subtypes of Nav channels and FHFs that are functional in raphe neurons. For example, we assumed here that these neurons express a kinetically uniform Nav channel population and all channels are equally interacting with the FHF. However, the 80% to 20% ratio between normal and long-term inactivation components may also be explained by two (or more) Nav populations (e.g., defined by channel subtype or compartmentalization), each interacting differently (or not at all) with the FHF, in a concentration-dependent manner. Although we are not aware of reports in the literature where the total $Na^+$ current exhibits significantly more than 20% LTI under physiological conditions, the LTI fraction can be higher in FHF-transfected neurons, where FHF may reach higher concentration (*Laezza et al., 2009*). Interestingly, the same type of kinetic model with open-state block can account for the resurgent $Na^+$ current (*Raman and Bean, 2001*). However, in that case, the entry and exit time constants have comparable values. That model can also be considered a leaky integrator but the fraction of channels that enter the blocked state during a brief depolarization becomes available immediately upon repolarization and thus augments the subthreshold depolarizing $Na^+$ current, helping the hosting neuron to spike faster. In contrast, LTI plays the opposite role, by decreasing the amount of $Na^+$ current (both sub- and suprathreshold) and making it harder for the neuron to spike at high frequency.

We are not aware of other studies where Nav channels – or other channels – have been specifically identified as molecular leaky integrators, even though the leaky integrator is a powerful concept that has long been associated with neural computation, from individual neurons and circuits to cognitive processes (*Knight, 1972*; *Cook and Maunsell, 2002*; *Mitani et al., 2013*; *Portugues et al., 2015*; *Groschner et al., 2018*; *Bahl and Engert, 2020*). In excitable cells, a molecular leaky integrator can be a tool for monitoring and regulating cellular activity, as we demonstrated here with Nav channels and LTI in serotonergic raphe neurons. Desensitization of nicotinic receptors is another example of a regulatory molecular process characterized by (relatively) fast onset in the presence of neurotransmitter and a phosphorylation-dependent mixture of fast and slow recovery in the absence of neurotransmitter (*Paradiso and Brehm, 1998*). Like Nav channels, potassium channels can also inactivate and recover from inactivation (or activate and deactivate) slowly, on time scales that are longer than the duration of a single action potential and thus can modulate neuronal activity (*Schwindt et al., 1988*; *Storm, 1988*; *Bond et al., 2005*; *Khaliq and Bean, 2008*; *Greene and Hoshi, 2017*), and can even implement a form of short-term memory that relies solely on intrinsic neuronal excitability properties (*Turrigiano et al., 1996*). More generally, activity-dependent slow inactivation and recovery from inactivation of Nav and other channels and receptors could be a mechanism that stabilizes cellular function against fluctuations in expression levels (*Ori et al., 2018*).

The intracellular $Ca^{2+}$ also acts as a leaky integrator, quickly incrementing its concentration with each action potential, via $Ca^{2+}$ influx through voltage-gated calcium channels, and more slowly decrementing it during periods of quiescence (*Gorman and Thomas, 1978*; *Helmchen et al., 1996*). Thus, the envelope of intracellular $Ca^{2+}$ concentration becomes a measure of cellular activity that can be used to regulate a variety of $Ca^{2+}$-dependent cellular processes (*Gárdos, 1958*; *Yuste et al., 2000*; *O'Leary et al., 2013*), including the activity of many types of ion channels (*Meech and Standen, 1975*; *Keen et al., 1999*; *Peterson et al., 1999*; *Deschênes et al., 2002*; *Wen and Levitan, 2002*; *Hartzell et al., 2005*), which in turn can change the firing activity of the cell. Enzymes such as CaMKII can also function as a leaky integrator, as they rapidly activate during episodic rises in $Ca^{2+}$, and then slowly deactivate as $Ca^{2+}$ decays (*Chang et al., 2017*). As a leaky integrator established by the long-term inactivation process, Nav channels add a powerful mechanism for sensing and regulating the activity of excitable cells.

## Materials and methods

All animal procedures were approved by the Animal Care and Use Committees of the University of Missouri and SUNY Downstate Health Sciences University.

### Brainstem slices

In vitro medullary slices containing RO neurons were obtained from neonatal (postnatal days 1 – 4) Sprague Dawley male and female rats (Charles River Laboratory Inc, USA, RRID:RGD_737891), as previously described *Koshiya and Smith (1999)*. Briefly, animals were anaesthetized with isoflurane and the brainstem was swiftly removed in artificial cerebral spinal fluid (aCSF) containing the following (in mM): 124 NaCl, 25 NaHCO$_3$, 3 KCl, 1.5 CaCl$_2$, 1 MgSO$_4$, 0.5 NaH$_2$PO$_4$, and 30 D-glucose, equilibrated with 95% O$_2$ and 5% CO$_2$ (pH 7.4 ± 0.05 at room temperature). Transverse slices (300 – 400 μm thick) containing *nucleus raphe obscurus*, the pre-Bötzinger Complex, and hypoglossal (XII) nerve rootlets were cut on a Campden Instruments 7000 vibratome (Campden Instruments, England) and transferred to the recording chamber and superfused with aCSF at room temperature, at a rate of ≈ 5 ml/min. Raphe neurons were generally identified based on their location in the slice, adjacent to the midline. Whenever possible, the neurons were further selected based on their spiking pattern: regular and slow pacemaking (3 – 5 Hz) present in cell-attached mode, and broad action potentials (3 – 6 ms) with prominent calcium shoulder (*Ptak et al., 2009*). Whole-cell patch-clamp was done under IR-Dodt contrast imaging, using a Hamamatsu Flash 4.0 camera (Hamamatsu Photonics, Japan) controlled by the QuB software (RRID:SCR_018076) (*Navarro et al., 2015*). All recordings were obtained at room temperature.

### Acutely dissociated neurons

Dorsal raphe neurons were acutely dissociated from adult male Sprague-Dawley rats (200 – 250 g), as previously described (*Penington et al., 1991*). Briefly, animals were anaesthetized with isoflurane and then decapitated with a small animal guillotine. A small volume of gray matter was cut from immediately below the cerebral aqueduct containing the dorsal raphe nuclei, chopped into pieces, and bathed for 2 hr at room temperature, in a PIPES buffer solution containing 0.2 mg/mL trypsin (Sigma Type XI) under pure oxygen. The tissue was then triturated in Dulbecco's modified Eagle's medium to free individual neurons. Small droplets containing suspended neurons were placed in the recording chamber and cells were allowed to settle and adhere to the bottom of the chamber. An extracellular recording solution containing (in mM): 120 NaCl, 10 TEACl, 20 HEPES, 30 sucrose, 3 KCl, 1.5 CaCl$_2$, 1 MgCl$_2$ (pH 7.4 ± 0.05 with CsOH at room temperature) was continuously perfused at a rate of ≈ 2 ml/min. Neurons with truncated dendrites and soma ≥20 μm were selected for whole-cell patch-clamp. All recordings were obtained at room temperature.

### Solutions

For voltage clamp (VC) in brain slices, pipettes were filled with a solution containing (in mM): 70 Cs-gluconate, 30 Na-gluconate, 10 tetraethylammonium-Cl (TEA-Cl), 5 4-aminopyridine (4-AP), 10 EGTA, 1 CaCl$_2$, 10 HEPES, 4 Mg-ATP, 0.3 Na$_3$-GTP, 10 Na$_2$-phosphocreatine, pH 7.4 with CsOH (285 ± 5 mOsm/L). For VC in dissociated neurons, the pipette solution contained (in mM): 90 Cs-gluconate, 30 NaCl, 10 TEA-Cl, 5 4-AP, 20 HEPES, 10 EGTA, 1 CaCl$_2$, 4 Mg-ATP, 0.3 Na$_3$-GTP, 10 Na$_2$-phosphocreatine, pH 7.4 with CsOH. Cs$^+$, TEA$^+$, and 4-AP minimized K$^+$ currents, whereas the elevated Na$^+$ concentration decreased the size of Na$^+$ currents and reduced VC artifacts. For current clamp (CC) and dynamic clamp (DC) in brain slices, pipettes were filled with a solution containing (in mM): 125 K-gluconate, 4 NaCl, 10 EGTA, 1 CaCl$_2$, 10 HEPES, 4 Mg-ATP, 0.3 Na$_3$-GTP, 4 Na$_2$-phosphocreatine, pH 7.4 adjusted with KOH (285 ± 5 mOsm/L). For VC in brain slices, CdCl$_2$ (200 μM) and 7-nitro-2,3-dioxo-1,4-dihydroquinoxaline-6-carbonitrile (CNQX; 20 μM) were added to the superfusing aCSF to block Ca$^{2+}$ currents and inhibit synaptic transmission. CdCl$_2$ (200 μM) was also used for VC in dissociated neurons, while CNQX (20 μM) was used for CC and DC experiments in brain slices. To block Na$^+$ currents in brain slices, tetrodotoxin (TTX, 1 μM) was added to the superfusing aCSF. All reagents were purchased from Millipore-Sigma (St. Louis, MO), with the exception of Cs-gluconate from Hello Bio Inc (Princeton, NJ) and TTX from Alomone Labs (Jerusalem, Israel).

## Electrophysiology

For brain slices, pipettes (5 – 7 MΩ) were pulled from borosilicate glass. For dissociated neurons, pipettes (2 – 3 MΩ) were pulled from soda-lime glass. All pipettes were coated with Sylgard to reduce capacitive transients. Pipette capacitance was compensated 100% in VC and ≈ 75% in CC and DC. For DC and offline analysis, membrane capacitance ($C_m$) was approximated as the value used for compensation in VC and was typically 20 pF. Series resistance ($R_s$) was typically 9 – 15 MΩ. Cells with $R_s$ > 15 MΩ or with evidence of poor space-clamp were discarded. In VC experiments, $R_s$ was compensated ≈ 80% and the compensation was readjusted before running a protocol. In CC and DC experiments, $R_s$ was compensated 100% and periodically readjusted. Measured liquid junction potentials of ≈ 10 mV for the $K^+$-based and ≈ 8 mV for the $Cs^+$-based solutions were corrected online. For neurons in the slice preparation, whole-cell recordings were obtained with an EPC-10 patch-clamp amplifier (HEKA Electronik, Germany), controlled by Patchmaster software (HEKA Elektronik, Germany, RRID:SCR_000034). Manipulators and stage motors (Scientifica, United Kingdom) were controlled with the QuB software for semi-automated cell targeting (*Navarro et al., 2015*). For VC, the recorded currents were low-pass filtered at 40 kHz and digitally sampled at 100 kHz. For CC and DC, the membrane voltage signal was digitally sampled at 50 kHz (open-bandwidth). For acutely dissociated neurons, whole-cell recordings were obtained with an Axopatch 200B patch-clamp amplifier (Molecular Devices), controlled by pClamp 10.3 software (Molecular Devices, RRID:SCR_011323). In this case, the recorded currents were low-pass filtered at 10 kHz and digitally sampled at 50 kHz.

## Voltage clamp experiments

Voltage clamp protocols (*Figure 2*) were constructed and applied with the Patchmaster program (brainstem slices) or with pClamp 10.3 (acutely dissociated neurons). The intersweep interval was 6 s at −80 mV, necessary for complete recovery from inactivation of $Na^+$ currents. Recordings with evidence of $Na^+$ current instability were discarded. Leak currents were subtracted using the P/n procedure. For VC recordings in brain slices, the TTX-sensitive $Na^+$ current was isolated via TTX subtraction.

## Nav kinetic model

For computer simulations and dynamic clamp experiments, we used the LTI model shown in *Figure 3A*, which is based on Model II from *Milescu et al. (2010b)*, and the non-LTI model shown in *Figure 1D*. Briefly, each rate constant has an Eyring expression defined as $k = k^0 \times e^{k^1 \times V}$, where $k^0$ [$ms^{-1}$] and $k^1$ [$mV^{-1}$] are pre-exponential and exponential factors, respectively, and $V$ is membrane potential. The rate constant values of the LTI model are given in '*Figure 3—source data 1*' file. The non-LTI model has the same rate constants but lacks the long-term inactivated state $S_{13}$ and the corresponding transition.

## Dynamic-clamp experiments

To inject Nav conductance in live neurons, under bath-applied TTX, we used the dynamic clamp functionality in the QuB software (*Milescu et al., 2008*), following previously described procedures (*Milescu et al., 2010b*). In the experiment described in *Figure 6*, the depolarizing current ($I_{Inj}$) was not normalized to cell capacitance, and values greater than 40 pA were not used, as they consistently resulted in depolarization block. The real-time computational loop was run at 50 kHz and the Nav model was solved using the matrix method. The software was run on a dual-processor workstation with Xeon E5-2667 v2 8-core CPUs, running Windows 7, interfaced with the patch-clamp amplifier via a National Instruments data acquisition board PCIe-6361 and BNC-2120 connector block.

## Computer simulations

To simulate the response of the model to voltage clamp protocols, we used the freely available MLab edition of the QuB software (http://www.milesculabs.org/QuB.html), as previously described *Milescu et al. (2008)*; *Milescu et al. (2010b)*.

## Statistical analysis

Data were analyzed statistically with Prism 4.1 (GraphPad, RRID:SCR_002798). In all cases, the sample size was sufficient for $\alpha = 0.05$ and a power of test of 0.8.

## Acknowledgements

We thank the members of the Milescu labs, Zeke Elkins, Dr. Kenneth Paradiso, and Dr. Philip Gottlieb for their input and comments on the manuscript, Dr. Andrew McClellan for his advice on electronic leaky integrators, and Dr. Florian Engert for his input on neural leaky integrators. We are very grateful to Dr. Kevin Cummings and Jenn Cornelius-Green for their assistance with the animal work, and to Dr. Sergei Sukharev for his generous support. This work was supported by American Heart Association grants 13SDG16990083 to LS Milescu and 13SDG14570024 to M Milescu, a training grant fellowship from the Graduate Assistance in Areas of National Need Initiative/Department of Education to MA Navarro, and Life Science Undergraduate Research Opportunity Program fellowships from the University of Missouri to JL Lin and LC Cowan. We dedicate this work to our late colleague, mentor, and friend, Dr. Troy Zars.

## Additional information

### Funding

| Funder | Grant reference number | Author |
|---|---|---|
| American Heart Association | 13SDG16990083 | Lorin S Milescu |
| American Heart Association | 13SDG14570024 | Mirela Milescu |
| U.S. Department of Education | Graduate Assistance in Areas of National Need Initiative Training Grant | Marco A Navarro |
| University of Missouri | Life Sciences Undergraduate Research Opportunity Program fellowship | Jenna L Lin Luke M Cowan |

The funders had no role in study design, data collection and interpretation, or the decision to submit the work for publication.

### Author contributions

Marco A Navarro, Conceptualization, Formal analysis, Validation, Investigation, Methodology, Writing - review and editing; Autoosa Salari, Conceptualization, Methodology, Writing - review and editing; Jenna L Lin, Luke M Cowan, Conceptualization, Investigation, Methodology, Writing - review and editing; Nicholas J Penington, Conceptualization, Resources, Data curation, Formal analysis, Validation, Investigation, Methodology, Writing - review and editing; Mirela Milescu, Conceptualization, Resources, Formal analysis, Supervision, Funding acquisition, Validation, Investigation, Methodology, Project administration, Writing - review and editing; Lorin S Milescu, Conceptualization, Resources, Data curation, Software, Formal analysis, Supervision, Funding acquisition, Validation, Investigation, Visualization, Methodology, Writing - original draft, Project administration, Writing - review and editing

### Author ORCIDs

Marco A Navarro https://orcid.org/0000-0002-2443-102X
Autoosa Salari https://orcid.org/0000-0002-8755-4553
Jenna L Lin http://orcid.org/0000-0001-9116-1763
Luke M Cowan https://orcid.org/0000-0002-5512-1227
Nicholas J Penington https://orcid.org/0000-0003-4153-0078
Mirela Milescu https://orcid.org/0000-0002-7152-2194
Lorin S Milescu https://orcid.org/0000-0002-3177-7010

## Ethics

Animal experimentation: All animal procedures followed the recommendations in the Guide for the Care and Use of Laboratory Animals of the National Institutes of Health, and were approved by the Institutional Animal Care and Use Committees of the University of Missouri (animal protocol #9397) and SUNY Downstate Health Sciences University (animal protocol #15-10479). All surgery was performed under anaesthesia with isoflurane, and every effort was made to minimize suffering.

## Decision letter and Author response

Decision letter https://doi.org/10.7554/eLife.54940.sa1
Author response https://doi.org/10.7554/eLife.54940.sa2

# Additional files

## Supplementary files

• Transparent reporting form

## Data availability

All data generated or analyzed during this study are included in the manuscript and in the source data files for Figures 2, 3, 5, and 6.

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
