## [Decision Letter]

**Acceptance summary:**

Several recent papers have described an interesting new mechanism of inactivation in voltage-activated sodium (Nav) channels whereby short depolarizations of a few ms cause a fraction of sodium channels to enter a state from which recovery from inactivation is very slow. However, the functional significance of this mechanism of long-term inactivation has not been explored extensively. Here, the authors present a nice combination of modeling and experiments (tied together by dynamic clamp experiments) to show long-term inactivation acts as a leaky integrator and that the reduction of sodium channel availability acts as negative feedback element to regulate pacemaking frequency of serotonergic neurons. The authors have done a nice job of addressing the concerns of the reviewers and the manuscript now makes a compelling demonstration that long-term inactivation, by slowing the recovery of Nav channels from inactivation, slows firing rate and places a limit on possible firing rates. The manuscript provides a well-reasoned presentation of the background information motivating the present work and a balanced discussion of how changes in firing may also be influenced by other conductances whose activation may be shaped by long-term inactivation.

**Decision letter after peer review:**

[Editors’ note: the authors submitted for reconsideration following the decision after peer review. What follows is the decision letter after the first round of review.]

Thank you for submitting your work entitled “Sodium channels implement a molecular leaky integrator that detects action potentials and regulates neuronal firing” for consideration by *eLife*. Your article has been reviewed by three peer reviewers, and the evaluation has been overseen by Kenton Swartz as Reviewing Editor and a Senior Editor. The following individuals involved in review of your submission have agreed to reveal their identity: Bruce P Bean (Reviewer #2); Arpad Mike (Reviewer #3).

Our decision has been reached after consultation between the reviewers. Based on these discussions and the individual reviews below, we regret to inform you that your work will not be considered further for publication in *eLife*.

Although the clear analysis and insightful discussion were appreciated by the reviewers, there are not a lot of new experimental results in the manuscript (mainly the data in Figure 5 showing how firing frequency depends on the level of sodium current availability) and many of the ideas have been touched on previously. Also, the reviewers found it odd that while the main point of the paper is proposing that long-term inactivation serves as a negative feedback element for regulating pacemaking frequency, you do not try to quantify this effect, for example by doing the dynamic clamp experiments in Figure 5 using models both with and without long-term-inactivation (i.e., with and without state S13). Although you will see in the reviews that there were diverging opinions, the consensus is that the work needs to be taken further to be appropriate for *eLife*. If you choose to experimentally show that a feedback mechanism involving LTI has a significant effect to control firing rate in either these cells or another types of neuron, we would be willing to reconsider the work as a new manuscript. *eLife* takes the position that we shouldn't require extensive new experiments for a revision, precisely because we believe the authors should decide what they wish their paper to be. That is why we have a two month revision time, and reject papers that might require extensive new work.

Reviewer #1:

This paper touches on the important question of how modulation of recovery from inactivation of sodium channels by fibroblast growth factor homologous factors (FHFs) may impact on cell excitability. Although this reader found the manuscript interesting to read, overall the paper seems to be a proof-in-principle that use-dependent changes in Nav availability can impact on cell firing frequency, with little in the way of hard proof that the models used are correct and that the available data are sufficient to adequately constrain any given model of “long-term-inactivation”. Although Figure 5 provides the punch line that spiking frequency is altered by Nav availability, it ends up essentially being a teaser, since the authors do not explicitly test how the presence or absence of LTI in their dynamic clamp models might differentially alter firing. This essentially leaves any reader hanging. Why were tests comparing the consequences of adding or removing LTI in the model not considered? Overall, although a proof-in-principle may have merit, it does not seem adequately developed here, or presented in a form that clearly delineates what the authors want to accomplish with this.

The authors' strong preference for a variety of metaphors (whether studying for an exam or home furnaces) to explain the phenomenology seems to brush aside the essential and long-understood point that use-dependent changes in conductances (whether reflecting intrinsic properties of channels, or slow modulatory changes in channels) are major contributors to regulation of cell excitability, whether it is M-current, SK current, or, as implied here, Nav inactivation. Why is “leaky integrator” a useful concept either for explaining the phenomenology to readers or for enabling understanding of the physiological role of long-term inactivation? This is not clear to this reader.

The structure of the paper also seemed a bit problematic. The paper seems to jump back and forth between data and model predictions and at times it seemed confusing whether the authors were discussing data or model behavior. Despite extensive use of a particular model, there is little in the paper to indicate how data constrain different potential models for the two pathways of inactivation, and there is little to address the extent to which the authors' preferred model is really constrained by data. Another issue is that the available data do not seem adequate to really discern the extent to LTI may or may not occur from closed states. This seems particularly problematic when the authors point out in subsection “Nav channels drive spiking frequency” that some of the differences seen in Figure 5 may arise from more closed-state inactivation, which is really poorly defined for both LTI or normal fast inactivation.

Figure 5. This is potentially the most interesting and exciting aspect of the paper, but as noted above no attempt to really exploit the advantages of manipulations of the Nav model in dynamic clamp were done. Furthermore, in regards to AP firing during a dynamic clamp experiment, changes in Nav density may also change AP duration, or peak amplitude such that there may be activity dependent changes in Cav currents or Ca^2+^-dependent currents, which may also impact on the firing properties. The result in Figure 5 is not sufficient on its own to assert that the changes with current density exclusively reflect Nav current. That may be the case, but other factors may also contribute.

As the authors mention, there is also the issue of what Nav currents are present in the raphe neurons and what FHFs may be present. In the absence of such information, it seems a bit premature to be building models, although perhaps as an explicit proof-in-principle that use-dependent reduction in Nav availability has certain effects might be appropriate. But in such a case, the explicit tests of adding back Nav currents with and without LTI would have seemed the minimum necessary.

Reviewer #2:

A number of recent papers have described a molecular basis of the phenomenon of long-term inactivation, a phenomenon by which short depolarizations of a few ms cause a fraction of sodium channels to enter a state from which recovery from inactivation is very slow. However, the functional significance of long-term inactivation has not been explored much. Here, the authors present a nice combination of modeling and experiment (tied together by dynamic clamp experiments) to show long-term inactivation acts as a leaky integrator and that the reduction of sodium channel availability acts as negative feedback element to regulate pacemaking frequency of serotonergic neurons. There are not a lot of new experimental results in the manuscript (mainly the data in Figure 5 showing how firing frequency depends on the level of sodium current availability) and some of the ideas have been touched on previously, but the clear analysis and insightful discussion seemed to me very useful and illuminating.

The authors propose that long-term inactivation serves as a negative feedback element for regulating pacemaking frequency, but they never try to quantify this effect. This could easily be done by doing the dynamic clamp experiments in Figure 5 using models both with and without long-term-inactivation (i.e., with and without state S13). In fact the point of Figure 5 was to show how pacemaking frequency depends on *G*_Na_, so it seemed a little strange that they used the model with S13 in which the feedback element is already present. It would seem more logical to first show the dependence without long-term inactivation and then show the effect of having the negative feedback element from long-term inactivation

Reviewer #3:

This manuscript demonstrates a novel function of sodium channels as cellular computational tools: beyond the well-known spike generator, and less general pacemaking driver functions, it introduces them as leaky integrators, i.e., a sort of a short-term memory device.

Serotonergic neurons of the raphe keep a memory of their past actions. Recent activity makes neurons less responsive, while recent inactivity makes them more responsive. The manuscript identifies the key element in this mechanism, the peculiar type of inactivation of sodium channels in raphe serotonergic neurons, termed long term inactivation (LTI). Similar to the rapid onset-rapid offset channel block that enables resurgent currents e.g., in cerebellar Purkinje neurons, this form of inactivation depends on certain channel-associated proteins (FHFs); however in this case the onset is fast, but the offset is slower by three orders of magnitude. Fast onset allows the sodium channel population to “remember” each action potential, because at each individual AP ~20% of the available channel population is trapped in LTI state. Slow recovery allows the channel population to “forget” with a time constant of several hundreds of milliseconds, thus ensuring that “memory” is not kept indefinitely.

The concept described in the manuscript is significant, experimental and computation methods are sound and reliable.

The manuscript is written in a very clear and concise style, which makes understanding easy. However, a few points should be made more emphatic:

– An operable mechanism requires a coding (frequency to available fraction) and a decoding (available fraction to frequency) process. At the beginning of the section “Nav channels drive spiking frequency” it should be clearly stated that the coding part is done, and authors start discussing the decoding part.

– Within the decoding part, there is a positive feedback mechanism that promotes firing: depolarization (over -53 mV) causes activation, activation causes further depolarization, etc., until the whole available population responds. Interestingly, there seems to be another positive feedback mechanism, which however hinders firing: slow depolarization favors closed state inactivation, which depletes available channel population, which slows down depolarization, etc. It would be interesting to discuss the nonlinearity caused by the two competing positive feedback mechanisms.

– In Figure 6 the controller mechanism of raphe serotonergic neurons is compared to a general controller. As far as I understand, the proportional and integral error correction has its parallel mechanism in the neurons. What about the derivative error correction?

– When discussing molecular leaky integrators, the recent description of CaMKII as such instrument (Chang et al., 2017) should be reviewed.

[Editors’ note: further revisions were suggested prior to acceptance, as described below.]

Thank you for resubmitting your work entitled “Sodium channels implement a molecular leaky integrator that detects action potentials and regulates neuronal firing” for further consideration by *eLife*. Your revised article has been evaluated by Kenton Swartz (Senior Editor and Reviewing Editor) and the original three reviewers.

The editor and reviewers think that the manuscript has been improved by the inclusion of new results and revisions to the text. However, the reviewers have identified a few remaining issues that need to be addressed before acceptance, as outlined below:

Reviewer #1:

This manuscript now makes a more compelling demonstration that LTI, by slowing the recovery of Nav channels from inactivation, slows firing rate and places a limit on possible firing rates. The manuscript provides a well-reasoned presentation of the background information motivating the present work and a balanced discussion of how changes in firing may also be influenced by other conductances whose activation is altered by LTI-dependent changes in AP properties. I have no major concerns about the manuscript, but list several issues upon which the authors may want to provide additional comment.

1) The general ratio of 80% fast recovery and 20% slow recovery argues that the rate of entry into standard fast recovery states is 4-fold faster than rate of entry into slow pathways, consistent with the rates in source data for Figure 3. Is this fast inactivation/LTI ratio similar to what is observed in other reports of LTI? For example, although the literature on FGFs does not generally use protocols that allow clear determination of fast and slow recovery, there do seem to be cases where the slow component of recovery is much more substantial than described here, implying that the rates of onset of fast inactivation and LTI are more comparable (perhaps Figure 3 in Laezza et al., 2009). Although the present paper does not address it, some cautionary remarks pertinent to factors that might impact on the ratio of tauf vs. tauLTI might be warranted in the Discussion. The present analysis assumes that all Nav channels are identical in these cells, in terms of molecular and functional properties of the underlying LTI process. Although there is no data here (or elsewhere about which this reader is aware) that addresses this, the 80%:20% ratio could arise from other factors, e.g., some channels containing a component necessary for LTI and other channels lacking it. That does not negate the principle point of the paper, which is the consequences of LTI. In fact, the relative importance of LTI would likely be greater in cells for which the fraction of slow recovery is greater.

2) Figure 5. The impact of including LTI is somewhat disappointingly small. In fact, by the measures shown here, the firing differences between inclusion of LTI or its absence would essentially be impossible to distinguishable by an observer asked to say whether a given firing behavior was consistent with the presence of LTI or not.

If one compares AP properties, peak, duration, and so on, are there differences between the LTI and non-LTI cases that might suggest that other conductances are also coming into play? In Figure 5B, the errors appear to SEM. Seems a bit surprising that anything less than 0.05 was actually obtained, when comparing the two cases.

Figure 5 should probably specific in the legend which raphe neurons were being used, and the same in Figure 6.

3) In Figure 6, shouldn't the plot in D at least up to 20 pA more closely parallel Figure 5B? In Figure 5B, the LTI and non-LTI are converged at 20 pA while in Figure 6 they are moving apart. Any explanations for this? It isn't clear why the differences in protocols would produce this, or is it simply variation among sets of cells?

Reviewer #2:

The addition of the modeling with and without LTI in Figure 5 and 6 fills the obvious gap in the manuscript, and I thought the revised manuscript makes an important contribution by nicely illustrating the functional role of LTI in these neurons.

I found it very difficult to understand the new paragraph four of the Discussion, particularly the statement “Lowering the firing frequency would extend the time spent by Nav channels above -80 mV, which would enhance closed-state inactivation.” This seems impossible to interpret without knowing why and how firing frequency is lowered. And it seems counter-intuitive as a general statement because in general a slow-firing cell would seem to spend less time not more at depolarized voltages. I guess the authors are specifically thinking of a cell where firing is slower because of a more depolarized spike threshold. At the least, the authors need to make this section easier to understand. Even if it were made clearer, I was not convinced it conveys an idea that is important to the paper.

Reviewer #3:

All concerns of the reviewers have been adequately addressed.

In Figure 5, I would include both LTI and non-LTI data in panels A and C as well.

---

## [Author Response]

[Editors’ note: the authors resubmitted a revised version of the paper for consideration. What follows is the authors’ response to the first round of review.]

Reviewer #1:This paper touches on the important question of how modulation of recovery from inactivation of sodium channels by fibroblast growth factor homologous factors (FHFs) may impact on cell excitability. Although this reader found the manuscript interesting to read, overall the paper seems to be a proof-in-principle that use-dependent changes in Nav availability can impact on cell firing frequency, with little in the way of hard proof that the models used are correct and that the available data are sufficient to adequately constrain any given model of “long-term-inactivation”. Although Figure 5 provides the punch line that spiking frequency is altered by Nav availability, it ends up essentially being a teaser, since the authors do not explicitly test how the presence or absence of LTI in their dynamic clamp models might differentially alter firing. This essentially leaves any reader hanging. Why were tests comparing the consequences of adding or removing LTI in the model not considered? Overall, although a proof-in-principle may have merit, it does not seem adequately developed here, or presented in a form that clearly delineates what the authors want to accomplish with this.

Thank you for your encouraging comments. As explained in the beginning of this document, we have now added the comparison between LTI and non-LTI models, showing how they differentially alter firing.

As far as models, we completely agree that a study should offer hard proof that models are well constrained by data. We therefore understand the reviewer's skepticism, but we think it may stem from a misunderstanding: the LTI Nav model used in this study is in fact very much grounded in experimental data, being originally formulated in the Milescu et al., 2010 paper (referenced multiple times throughout the manuscript). There, we derived a Nav state model from a comprehensive collection of data that covered steady-state properties (activation and inactivation), as well as kinetic properties (time course of activation and inactivation, slow inactivation, recovery from inactivation, closed state inactivation, entry into LTI, use-dependence, etc.), all at multiple voltages and time scales (tens of microseconds to seconds). The model was then tested in ventral raphé neurons through a series of dynamic clamp experiments and verified that it is capable of producing spiking with properties very similar to what the endogenous sodium current produces. For modeling, we used the constraining algorithms developed by L.M. and implemented in the QuB software (Navarro et al. and Salari et al., 2018). For dynamic clamp, we also used the algorithms developed in QuB (Milescu et al., 2008). Overall, we think the model is really well constrained and we used it in this study.

We clarified these points in the manuscript (subsection “Mechanistic consequences of long-term inactivation”).

The authors' strong preference for a variety of metaphors (whether studying for an exam or home furnaces) to explain the phenomenology seems to brush aside the essential and long-understood point that use-dependent changes in conductances (whether reflecting intrinsic properties of channels, or slow modulatory changes in channels) are major contributors to regulation of cell excitability, whether it is M-current, SK current, or, as implied here, Nav inactivation. Why is “leaky integrator” a useful concept either for explaining the phenomenology to readers or for enabling understanding of the physiological role of long-term inactivation? This is not clear to this reader.

We had no intention of brushing aside what is already known about ion channels as regulators of cellular excitability, which is a vast field, but our study is focused on sodium channels and thus most of our references cover this area. The revised manuscript includes additional references in the Discussion.

As far as metaphors, we tried to write the manuscript for the broad audience of *eLife*, and we think that metaphors, analogies, and examples help. Indeed, to explain how the controller works, we used the home furnace as something most everyone is familiar with and can immediately grasp. Nevertheless, in addition to metaphors, we also explained – mechanistically – the role of Nav channels as a controller of excitability, making the following points: (i) Nav LTI quantitatively approximates a leaky integrator; (ii) a leaky integrator can convert frequency into amplitude, thus Nav channels can “measure” the firing rate; (iii) the effectively available Nav conductance can directly influence spiking frequency; (iv) through these two reciprocal relationships (ii and iii), Nav channels act as a controller, not alone but in synergy with other cellular mechanisms that altogether appear to maintain a low firing rate against excitatory input. This mechanism has been tested directly, by comparing – in the same cell – the effects of LTI and non-LTI models. We hope these points are now clearer in the revised manuscript.

We think that “use-dependence” merely describes the observed phenomenology, whereas “leaky integration” actually explains the underlying mechanism in a mathematical and computational sense, and having a mechanism is a prerequisite for making quantitative predictions. We find it fascinating that an ion channel, through this simple interaction with auxiliary factors, implements such a powerful and ubiquitous computational operation, and this is what drives this manuscript.

The structure of the paper also seemed a bit problematic. The paper seems to jump back and forth between data and model predictions and at times it seemed confusing whether the authors were discussing data or model behavior.

We hope that the revised manuscript addresses this concern. If not, we would be grateful if the reviewer could give us more concrete examples, and we would be happy to make further changes.

Despite extensive use of a particular model, there is little in the paper to indicate how data constrain different potential models for the two pathways of inactivation, and there is little to address the extent to which the authors' preferred model is really constrained by data.

Please see the above discussion on how well the model is constrained by data. Please also note that the same conceptual model, with one non-conducting state off the open state, has been arrived at by other investigators. As with any aspect of a channel mechanism, it is very likely that the real kinetic scheme is more complex, but our model does a reasonable job at capturing the data, particularly considering the mix of fast and slow time scales that make the acquisition of good data quite difficult. Also, the model aligns reasonably well with biophysical and structural data for Nav channels.

Another issue is that the available data do not seem adequate to really discern the extent to LTI may or may not occur from closed states. This seems particularly problematic when the authors point out in subsection “Nav channels drive spiking frequency” that some of the differences seen in Figure 5 may arise from more closed-state inactivation, which is really poorly defined for both LTI or normal fast inactivation.

Our previous study (Milescu et al., 2010) addresses in extenso the properties of the model, including closed-state inactivation (q.v. Figure 2D). In principle, LTI is open-state block and shouldn't occur from closed states. However, we agree with the reviewer that the existence of closed-state LTI has not been extensively investigated, by us or others. Nevertheless, even if there were some closed-state LTI, the phenomenology described in Figures 1 and 2 in our manuscript remains valid and Nav channels would still be described as a leaky integrator.

As far as what we wrote, we tried to explain why the peak INa is disproportionately smaller when spiking is slower, which is probably a result of closed-state inactivation (regular, not LTI). In contrast, there is less time for closed state inactivation at higher frequencies, although the voltage envelope slightly increases and drives more channels into inactivation, as predicted by the steady-state inactivation properties (the channel closed-state inactivates above -80 mV).

We added a new paragraph in Discussion (paragraph four) that elaborates on the positive feedback created by closed-state inactivation.

Figure 5. This is potentially the most interesting and exciting aspect of the paper, but as noted above no attempt to really exploit the advantages of manipulations of the Nav model in dynamic clamp were done.

Thank you for your encouraging comment. We understand and have added the requested experiment. The new results are, indeed, exciting

Furthermore, in regards to AP firing during a dynamic clamp experiment, changes in Nav density may also change AP duration, or peak amplitude such that there may be activity dependent changes in Cav currents or Ca^2+^-dependent currents, which may also impact on the firing properties. The result in Figure 5 is not sufficient on its own to assert that the changes with current density exclusively reflect Nav current. That may be the case, but other factors may also contribute.

An excellent observation, and we already wrote in the initial submission that Nav channels may alter the firing not just directly, as a source of depolarizing current in the interspike interval, but also indirectly, by altering the shape of the AP and the ensuing currents that in turn change frequency. Whether the effects are direct or indirect, they are nonetheless caused by manipulating *G*_Na_, as shown in Figure 5. Understanding how other currents are involved would, of course, be very interesting but outside the scope of this study.

As the authors mention, there is also the issue of what Nav currents are present in the raphe neurons and what FHFs may be present. In the absence of such information, it seems a bit premature to be building models, although perhaps as an explicit proof-in-principle that use-dependent reduction in Nav availability has certain effects might be appropriate. But in such a case, the explicit tests of adding back Nav currents with and without LTI would have seemed the minimum necessary.

We agree with the reviewer that it would be interesting to know the identity of these molecular players in raphé neurons, but to fully understand the functional interactions would be a project in itself. The Venkatesan et al. reference suggested by the 2nd reviewer is a good example in this sense. Moreover, as the reviewer points out, the validity of the model and the proposed mechanism would not change once this information becomes available. Whether we know the type of FHF and the type of Nav, the Nav kinetic model would be obtained in exactly the same way, from the same data. A classical and very relevant example is the resurgent current, which was discovered and modeled (Raman and Bean, 1997) before its molecular identity was understood (Grieco et al., 2005).

We have now done those tests with LTI and non-LTI models.

Reviewer #2:[…]The authors propose that long-term inactivation serves as a negative feedback element for regulating pacemaking frequency, but they never try to quantify this effect. This could easily be done by doing the dynamic clamp experiments in Figure 5 using models both with and without long-term-inactivation (i.e., with and without state S13). In fact the point of Figure 5 was to show how pacemaking frequency depends on G_Na_, so it seemed a little strange that they used the model with S13 in which the feedback element is already present. It would seem more logical to first show the dependence without long-term inactivation and then show the effect of having the negative feedback element from long-term inactivation

Thank you for your suggestions. Yes, it may seem strange that we have not done the comparison LTI vs. non-LTI in the first place, but our intention was to present it in a different paper and focus the present manuscript on the leaky integrator concept, for a broader audience.

We have now done these experiments and the new data are presented in Figures 5 and 6.

Reviewer #3:[…].The manuscript is written in a very clear and concise style, which makes understanding easy. However, a few points should be made more emphatic:– An operable mechanism requires a coding (frequency to available fraction) and a decoding (available fraction to frequency) process. At the beginning of the section “Nav channels drive spiking frequency” it should be clearly stated that the coding part is done, and authors start discussing the decoding part.

We had already mentioned in the text that Nav channels "encode" frequency, but we liked your suggestion about "coding"/"decoding" and incorporated it in Results (subsection “Nav channels drive spiking frequency”).

– Within the decoding part, there is a positive feedback mechanism that promotes firing: depolarization (over -53 mV) causes activation, activation causes further depolarization, etc., until the whole available population responds. Interestingly, there seems to be another positive feedback mechanism, which however hinders firing: slow depolarization favors closed state inactivation, which depletes available channel population, which slows down depolarization, etc. It would be interesting to discuss the nonlinearity caused by the two competing positive feedback mechanisms.

This is a very good point and we added a paragraph in the Discussion (paragraph four) where we discuss two additional forms of (positive) feedback: voltage-dependent activation and closed state inactivation.

– In Figure 6 the controller mechanism of raphe serotonergic neurons is compared to a general controller. As far as I understand, the proportional and integral error correction has its parallel mechanism in the neurons. What about the derivative error correction?

It would be quite interesting if there were a correspondence between each of the three components (P, I, D) of the error correction mechanism in the general controller and a certain aspect of the Nav/neuronal controller. However, the Nav/neuronal controller is an extremely simple physical implementation of the general controller principle, and may not support the full functionality. In fact, the three error correction mechanisms are not all necessary for a controller to work – the proportional correction is sufficient. At the moment, we are not sure what types of correction are implemented through the LTI mechanism, but it's certainly something worth investigating.

We added a note in the revised manuscript (subsection “Nav channels are a molecular controller that regulates neuronal firing rate”).

– When discussing molecular leaky integrators, the recent description of CaMKII as such instrument (Chang et al., 2017) should be reviewed.

Thank you, we added the reference.

[Editors’ note: what follows is the authors’ response to the second round of review.]

The editor and reviewers think that the manuscript has been improved by the inclusion of new results and revisions to the text. However, the reviewers have identified a few remaining issues that need to be addressed before acceptance, as outlined below:Reviewer #1:This manuscript now makes a more compelling demonstration that LTI, by slowing the recovery of Nav channels from inactivation, slows firing rate and places a limit on possible firing rates. The manuscript provides a well-reasoned presentation of the background information motivating the present work and a balanced discussion of how changes in firing may also be influenced by other conductances whose activation is altered by LTI-dependent changes in AP properties. I have no major concerns about the manuscript, but list several issues upon which the authors may want to provide additional comment.1) The general ratio of 80% fast recovery and 20% slow recovery argues that the rate of entry into standard fast recovery states is 4-fold faster than rate of entry into slow pathways, consistent with the rates in source data for Figure 3. Is this fast inactivation/LTI ratio similar to what is observed in other reports of LTI? For example, although the literature on FGFs does not generally use protocols that allow clear determination of fast and slow recovery, there do seem to be cases where the slow component of recovery is much more substantial than described here, implying that the rates of onset of fast inactivation and LTI are more comparable (perhaps Figure 3 in Laezza et al., 2009). Although the present paper does not address it, some cautionary remarks pertinent to factors that might impact on the ratio of tauf vs. tauLTI might be warranted in the Discussion. The present analysis assumes that all Nav channels are identical in these cells, in terms of molecular and functional properties of the underlying LTI process. Although there is no data here (or elsewhere about which this reader is aware) that addresses this, the 80%:20% ratio could arise from other factors, e.g., some channels containing a component necessary for LTI and other channels lacking it. That does not negate the principle point of the paper, which is the consequences of LTI. In fact, the relative importance of LTI would likely be greater in cells for which the fraction of slow recovery is greater.

We agree with the reviewer that these matters are important and deserve further investigation. Indeed, we assumed here that Nav channels are identical and they all interact with the putative FHF in such a way as to produce the observed 80%:20% ratio between normal and long-term inactivation. However, as the reviewer notes, it could very well be two (or more) Nav populations, each interacting differently (or not at all) with the FHF.

We are also not aware of reports in the literature where the slow component of the total current is significantly greater than 20%, in wild-type neurons under physiological conditions. In Figure 3 in Laezza et al. the slow component appears to be <= 20% for all datasets except the data obtained from cells transfected with FGF14-1a.

Regarding the rates, we are not sure that the ratio of 80%:20% between normal and long-term inactivation components should be interpreted as meaning a ratio of 4:1 between the two corresponding entry time constants. It is possible to have a kinetic model with two exponential components that have similar time constants but different amplitudes, as shown in Figure 3B, where we simulated the time course of entry into normal (P_I_) and long-term (P_S_) inactivation states. As the figure indicates, the two entry time constants are very much similar and both components are complete within 2 ms, yet the state occupancies become 0.8 and 0.2, respectively.

We added a brief discussion of this (paragraph six, Discussion), including the Laezza et al. reference.

2) Figure 5. The impact of including LTI is somewhat disappointingly small. In fact, by the measures shown here, the firing differences between inclusion of LTI or its absence would essentially be impossible to distinguishable by an observer asked to say whether a given firing behavior was consistent with the presence of LTI or not.If one compares AP properties, peak, duration, and so on, are there differences between the LTI and non-LTI cases that might suggest that other conductances are also coming into play? In Figure 5B, the errors appear to SEM. Seems a bit surprising that anything less than 0.05 was actually obtained, when comparing the two cases.Figure 5 should probably specific in the legend which raphe neurons were being used, and the same in Figure 6.

We mentioned in the manuscript that there are other regulatory mechanisms besides LTI in raphe neurons, which in the absence of LTI still produce a good amount of adaptation. See, for example, the decay in frequency visible in Figure 6C, in the black traces obtained with the non-LTI model. Thus, the results of swapping LTI and non-LTI Nav models in the same cell are to some extent masked by these other mechanisms. If a cell had no other adaptation mechanism besides Nav LTI, the contrast between LTI and non-LTI models would be greater. Clearly, these neurons are trying hard to resist depolarizing input. Due to these other (Nav-unrelated) mechanisms, one would probably not be able to infer the presence of LTI in a cell from observing the adaptation in frequency alone. However, it has been shown in a previous study (Milescu et al., 2010) that there is a significant difference in how the AP shape changes in response to sustained depolarizations: with the LTI model, the AP broadens on both sides, whereas with the non-LTI model the AP broadens only on the downward side (Figure 7B in that paper).

We expanded the explanation at the end of subsection “Firing rate is lower in the presence of LTI”.

The test comparing the two data sets (non-LTI vs LTI) gives P~0.04, which is not too far from 0.05.

DR neurons were only tested for the presence of LTI. All the other experiments were done in RO neurons. We updated figure legends (where applicable) to include information on neuronal type and preparation.

3) In Figure 6, shouldn't the plot in D at least up to 20 pA more closely parallel Figure 5B? In Figure 5B, the LTI and non-LTI are converged at 20 pA while in Figure 6 they are moving apart. Any explanations for this? It isn't clear why the differences in protocols would produce this, or is it simply variation among sets of cells?

Figures 5B and 6D show different types of data: f vs *G*_Na_ in Figure 5B, obtained without stimulation (I_Inj_ = 0) and approaching convergence at 20 nS/pF (not 20 pA), and f vs I_Inj_ in Figure 6D, obtained with stimulation (I_Inj_ = 0…40 pA) and with *G*_Na_ = 10 or 15 nS/pF.

The only data points that can be compared between the two figures are:

– in Figure 5B: for G_Na_ = 10 nS/pF and I_Inj_ = 0, f ~4.7 ± 0.72 for LTI (black symbols) and f ~5.9 ± 0.72 for non-LTI (grey symbols);

– in Figure 6D: for I_Inj_ = 0 and G_Na_ = 10 nS/pF, f ~4.5 ± 0.4 for LTI (red symbols) and f ~5.3 ± 0.45 for non-LTI (black symbols).

The two sets of data presented in Figures 5 and 6 were obtained with different protocols and in different cells, hence the data points mentioned above are statistically comparable but not identical.

Reviewer #2:The addition of the modeling with and without LTI in Figure 5 and 6 fills the obvious gap in the manuscript, and I thought the revised manuscript makes an important contribution by nicely illustrating the functional role of LTI in these neurons.I found it very difficult to understand the new paragraph four of the Discussion, particularly the statement “Lowering the firing frequency would extend the time spent by Nav channels above -80 mV, which would enhance closed-state inactivation.” This seems impossible to interpret without knowing why and how firing frequency is lowered. And it seems counter-intuitive as a general statement because in general a slow-firing cell would seem to spend less time not more at depolarized voltages. I guess the authors are specifically thinking of a cell where firing is slower because of a more depolarized spike threshold. At the least, the authors need to make this section easier to understand. Even if it were made clearer, I was not convinced it conveys an idea that is important to the paper.

The other two reviewers suggested in the first submission that we comment on closed-state inactivation. We reformulated paragraph four of the Discussion to make it clearer.

Reviewer #3:All concerns of the reviewers have been adequately addressed.In Figure 5, I would include both LTI and non-LTI data in panels A and C as well.

Done.